# Doodle to Detect: A Goofy but Powerful Approach to Skeleton-based Hand Gesture Recognition

**SangHoon Han**[1,†]  **Seonho Lee**[1,†],  **Hyeok Nam**[1,†],  **JaeHyeon Park**[2],
**MinHee Cha**[2],  **MinGeol Kim**[1],  **HyunSe Lee**[2],  **SangYeon Ahn**[2],
**MoonJu Chae**[2],  **Sung In Cho**[1*]

[1] Department of Artificial Intelligence at Sogang University
35, Baekbeom-ro, Mapo-gu, Seoul 04107, Republic of Korea
{leo4102, sunhozizi, skagur10, rlaalsrjf6, csi2267}@sogang.ac.kr

[2] Department of Computer Science and Artificial Intelligence at Dongguk University
30, Pildong-ro 1-gil, Jung-gu, Seoul 04620, Republic of Korea
{pjh0011, rninicha, sae4394, sy990607, fulldoor}@dgu.ac.kr

## Abstract

Skeleton-based hand gesture recognition plays a crucial role in enabling intuitive human–computer interaction. Traditional methods have primarily relied on hand-crafted features—such as distances between joints or positional changes across frames—to alleviate issues from viewpoint variation or body proportion differences. However, these hand-crafted features often fail to capture the full spatio-temporal information in raw skeleton data, exhibit poor interpretability, and depend heavily on dataset-specific preprocessing, limiting generalization. In addition, normalization strategies in traditional methods, which rely on training data, can introduce domain gaps between training and testing environments, further hindering robustness in diverse real-world settings. To overcome these challenges, we exclude traditional hand-crafted features and propose Skeleton Kinematics Extraction Through Coordinated grapH (SKETCH), a novel framework that directly utilizes raw four-dimensional (time, $x$, $y$, and $z$) skeleton sequences and transforms them into intuitive visual graph representations. The proposed framework incorporates a novel learnable Dynamic Range Embedding (DRE) to preserve axis-wise motion magnitudes lost during normalization and visual graph representations, enabling richer and more discriminative feature learning. This approach produces a graph image that richly captures the raw data's inherent information and provides interpretable visual attention cues. Furthermore, SKETCH applies independent min–max normalization on fixed-length temporal windows in real time, mitigating degradation from absolute coordinate fluctuations caused by varying sensor viewpoints or differences in individual body proportions. Through these designs, our approach becomes inherently topology-agnostic, avoiding fragile dependencies on dataset- or sensor-specific skeleton definitions. By leveraging pre-trained vision backbones, SKETCH achieves efficient convergence and superior recognition accuracy. Experimental results on SHREC'19 and SHREC'22 benchmarks show that it outperforms state-of-the-art methods in both robustness and generalization, establishing a new paradigm for skeleton-based hand gesture recognition. The code is available at https://github.com/capableofanything/SKETCH.

---

*Corresponding author. † These authors contributed equally.

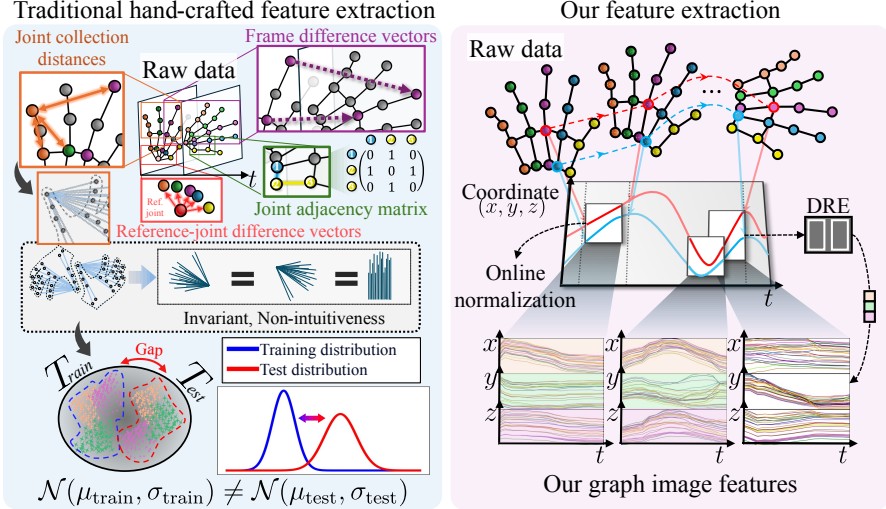

Figure 1: The left shows an example of the conventional methods dependent on the training data. The right shows the proposed feature extraction approach, which converts raw data into a visual representation and performs normalization based on real-time input data, with Dynamic Range Embedding.

# 1 Introduction

Hand gesture recognition using skeleton data has emerged as a key component in realizing sophisticated human–computer interaction across various fields, including intelligent robotics, mixed reality (MR), and human–robot interaction [21]. Hand gesture recognition via skeletal tracking contributes to the implementation of intuitive interfaces in virtual reality/augmented reality (VR/AR) environments and greatly assists robots in perceiving and interpreting human activities [7, 31]. In such environments, various sensors—such as RGB cameras, depth sensors, and LiDAR—capture human motion and the skeleton data extracted from these sensors (i.e., the 2D/3D coordinates of key joints) provides a concise representation of human posture that is less affected by changes in illumination. Unlike RGB video frames [15], skeleton data focuses on the structure of the body [19], offering higher processing efficiency and robustness against variations in background and lighting conditions. However, directly utilizing such raw skeleton data entails several inherent challenges for reliable gesture understanding in real-world environments.

This challenge is further compounded by the "geometric shortcut" phenomenon, as identified in Sonata [50], where directly inputting raw coordinate data can lead models to overfit to low-level spatial patterns—such as simply memorizing hand-height values—rather than extracting high-level features, ultimately resulting in representation collapse [6, 52]. Furthermore, raw skeletal coordinates are highly sensitive to sensor viewpoint and subject position, leading to substantial variation in observed sequences for identical actions. As a result, models trained under fixed-viewpoint assumptions may suffer degraded recognition accuracy during testing due to discrepancies in absolute coordinates caused by viewpoint shifts [23, 30]. In addition to these viewpoint-related issues [16], further variations may arise from individual-specific factors, resulting in distribution shifts between training and testing data. Such factors include differences in users' physical proportions (e.g., hand size and finger proportions) and individual variations in movement style and execution speed, all of which can significantly affect the resulting skeleton data.

To address these issues, conventional skeleton-based action recognition approaches [32, 33] generally do not primarily rely on using raw joint data directly (although there have been prior attempts to do so, their generalization performance is limited by the geometric shortcut; see Appendix A for details); instead, they focus on hand-crafted features designed to alleviate variations, which are extracted via rule-based methods from the raw data. As shown in Fig. 1, representations such as joint collection distances [8, 54], frame difference vectors [8, 60], joint adjacency matrices [60, 24], and reference-joint difference vectors [60, 24] are used to encode skeletal motion in time series through

rule-based processing, which directly extracts high-level features to mitigate geometric shortcuts and avoid issues related to absolute coordinate variability caused by differing viewpoints. Among them, adjacency matrices are employed to explicitly represent the structural relationships among joints for topology modeling, and the reference-joint difference vectors encode each joint in a sequence as a vector relative to a designated reference joint defined in the first frame.

However, existing hand-crafted feature extraction methods share common drawbacks: they are heavily dependent on domain expert knowledge, often lack interpretability, and struggle to fully capture the rich spatio-temporal information contained in raw skeleton data. Among them, methods based on reference-joint difference vectors suffer from additional inherent limitations. Their performance is highly sensitive to the choice of reference joint, requiring prior topology information. In practice, the same anatomical joint may be indexed differently across datasets, or may not be defined at all, leading to reliance on dataset- or sensor-specific topology definitions and limiting broader applicability. Furthermore, even with an optimal reference joint, these approaches remain fragile, as missing or noisy joints in real-world scenarios can severely degrade generalization performance, resulting in strong reference-joint dependency. Beyond these limitations, subject-specific variations such as differences in body proportions or motion styles remain unresolved. Due to the inherent coordinate variations caused by differences in subjects' physical proportions, the hand-crafted features extracted from raw data inevitably retain these variations. This necessitates normalization based on global statistics computed from the entire training dataset to reduce distribution shifts arising from subject-specific body scales. Notably, window-wise (i.e., fixed-length segments-wise), per-axis normalization is generally avoided in this context, as it may distort the extracted features and undermine the intended viewpoint-invariance mitigation of the representation. However global normalization can suppress fine-grained differences in joint movements due to the influence of joint coordinates with large variance, and it inherently assumes that the test data distribution will closely resemble that of the training set—an assumption often violated in real-world scenarios, potentially degrading model performance at inference time.

To overcome these limitations, this study—drawing inspiration from ViTST [22]—proposes a novel hand gesture recognition framework termed Skeleton Kinematics Extraction Through Coordinated grapH (SKETCH). Unlike conventional hand-crafted approaches, as shown in Fig. 1, the proposed SKETCH directly utilizes four-dimensional skeleton coordinate data, thereby eliminating the reliance on domain expert knowledge, enhancing interpretability, and preserving the rich information inherent in the raw data. To further address the domain gap between training and testing data caused by differences in subject-specific variations—issues that even hand-crafted features struggle to mitigate—SKETCH employs a window-based axis-wise min–max normalization strategy (see Appendix E.4 for ablation analysis). However, normalization alone is insufficient to resolve the inherent limitations of raw coordinate inputs, particularly the "geometric shortcut" phenomenon (as further detailed in Appendix A). To tackle this, we leverage a plotting technique as an intermediate medium to transform normalized coordinates into a graph image representation that alleviates viewpoint and subject variations, enabling more robust and semantically meaningful feature learning. Beyond its role in feature extraction, this design is inherently topology-agnostic, allowing the proposed SKETCH method to generalize across diverse skeleton structures and remain resilient to pose estimation errors (see Appendix E.6). The resulting visual graph also enhances interpretability by making it possible to analyze which spatiotemporal patterns or movement cues the model focuses on. This image-based representation further enables the use of backbones—pretrained on image classification datasets such as ImageNet-1k [37] and -22k [10]—to extract high-level features. By leveraging a pretrained backbone, our approach converges in fewer epochs for training than existing methods and achieves higher hand gesture classification accuracy. While the proposed window-based axis-wise min–max normalization effectively reduces domain shifts, it inevitably normalizes all coordinate axes to the same $[0, 1]$ range, resulting in the loss of relative movement scale across axes. To address this issue, we introduce Dynamic Range Embedding (DRE), which compressively encodes the movement range or variability of each axis into a scalar and injects it into the corresponding partition of the image. Consequently, the SKETCH framework—combining visual representation construction, window-based axis-wise normalization, and axis-wise DRE—enables robust recognition even under varying absolute scales or sensor viewpoints [23]. Furthermore, since each windowed input unit is processed independently, the model exhibits strong adaptation to environmental variability. This design is particularly well-suited for online (streaming) methods that process data on a per-frame basis for action recognition.

The main contributions of this study are as follows:

- **Direct utilization of raw data**: Uses raw joint coordinates without complex preprocessing, preventing information loss due to hand-crafted feature extraction and simplifying the data pre-processing pipeline.

- **A simple and intuitive image graphing method—Skeleton Kinematics Extraction Through Coordinated grapH (SKETCH)**: SKETCH is a novel plotting technique that converts time-series skeleton data into images, effectively mitigating the geometric short-cut problem [50], where models tend to overfit to low-level spatial cues when using raw coordinates. By applying window-wise per-axis min–max normalization and incorporating Dynamic Range Embedding (DRE) to encode each axis's movement range and variability, the method achieves stable real-time recognition despite variations in viewpoint, scale, user body proportions, and domain gaps between training and testing data. It is also inherently topology-agnostic, enabling robust generalization across diverse skeleton structures and real-world conditions.

- **Integration with vision models and interpretable image features**: By combining with powerful vision models such as ViT [11] and Swin Transformer [27], the proposed approach attains superior classification and detection accuracy compared to conventional methods and introduces a new paradigm—distinct from existing approaches—as an interpretable feature in the field of skeleton-based hand gesture recognition.

## 2 Related Work

Skeleton-based hand gesture recognition techniques primarily utilize sequence data composed of coordinates and frames. Depending on the characteristics of the dataset, existing methods can be broadly categorized into offline approaches [9] and online approaches [2, 4, 12].

### 2.1 Offline Classification Methods

As shown in Fig. 10(a), offline datasets are annotated such that each sequence corresponds to a single gesture class. Offline hand gesture recognition methods can be broadly divided into two approaches. First, CNN and LSTM-based methods [1, 25, 46, 47] utilize the current frame along with adjacent frame information to recognize gestures. This approach is effective for data with short sequence lengths but struggles to fully exploit information from frames that are temporally distant. Second, to simultaneously consider fine-grained gesture details and temporal information beyond what the first approach offers, recent studies have actively investigated architecture incorporating structures such as GCN [17] and Transformer [45].

CNN-based methods in [25, 46, 47] voxelize the joints and then apply a 3D CNN to simultaneously learn spatial and temporal information. DDNet [54] extracts joint collection distances (JCD) and employs a 1D CNN architecture to incorporate temporal information between frames for gesture classification. However, these CNN-based techniques [25, 54] do not define the relationships between joints, making it difficult to fully capture the geometric characteristics of joint configurations. Due to the issue of not defining the relationships between joints, GCN-based methods [17, 43, 53, 60] that consider joint connectivity have recently been actively studied. Notably, methods such as ST-GCN [53] and MS-ISTGCN [43] reflect the structural characteristics of the skeleton in a graph, modeling the interactions between adjacent joints using techniques like an adjacency matrix. However, since GCN-based approaches rely on connected adjacent information, they struggle to capture correlations between frames that are temporally distant. To address this, methodologies applying Transformer architectures—which can leverage long sequence time-series data—have also been proposed [13, 57]. Nevertheless, applying Transformer architecture requires a large amount of training data and extensive training [48, 49, 57, 58].

Offline recognition methods achieve high classification accuracy by generating a single prediction for an entire sequence labeled with one gesture class. However, they are unsuitable for real-world applications requiring frame-wise recognition in continuous input streams.

### 2.2 Continuous (Online) Classification

In real-world environments, hand gesture recognition is implemented using online hand gesture recognition methods as shown in Fig. 10(b). A segment is defined as a unit consisting of consecutive

frames that share the same label. In practical settings, gesture recognition results must be predicted at every frame from a streaming data sequence. Moreover, the input sequence contains multiple types of gestures, with the gesture's start and end points as well as intervening non-gesture segments. Therefore, online classification techniques must simultaneously perform the following two tasks: (1) determining the type of gesture at each frame, and (2) detecting the start and end points of gestures. Online recognition methods [18, 43, 57] can be broadly categorized into two approaches: two-stage and one-stage methods. Two-stage methods [18] first determine whether a gesture is present within a segment and then decide the gesture class for that segment. This approach has the drawback that its overall performance heavily depends on the effectiveness of the gesture presence detector. On the other hand, one-stage methods receive real-time input data and simultaneously predict the gesture's start frame and end frame as well as the gesture class. RNN-based models [20, 30] tend to forget past frames when the sequence length becomes long; to mitigate this, LSTM-based models [5, 36] have been proposed. However, since these methods classify frames based on sequential processing, parallel processing is challenging and training speeds are slow. Transformer-based models [57] enable parallel processing, but they face difficulties in achieving high performance when there is insufficient training data.

To overcome these issues, OO-dMVMT [8], which utilizes a sliding window technique, has been proposed and provides state-of-the-art (SOTA) performance. OO-dMVMT integrates spatial and temporal information by employing features such as joint collection distances (JCD) and the difference vectors for the same joint between consecutive frames. However, since the conversion from coordinates to JCD relies on the Euclidean distance between joints, some of the fine-grained spatial information contained in the raw coordinate data is lost. Moreover, the normalization statistics computed from the training dataset are also applied during testing, based on the assumption that the test data follows a similar distribution—an assumption that may not hold in real-world scenarios.

## 2.3 ViTST

To handle irregularly sampled time-series data, ViTST [22] converts time-series data into line graph images and then employs a pretrained vision transformer for classification. Specifically, each feature within the time-series data is represented as an individual line graph. In each line graph, the horizontal axis represents time, while the vertical axis denotes the observed values of the corresponding feature. The converted image is then fed into a vision transformer. When ViT is used as the backbone, the image representing the graph is divided into fixed-size patches, and global relationships are captured by comparing the similarities between these patches. In contrast, the Swin Transformer divides the input graph image into local Swin windows, performs self-attention within each Swin window, and adopts a hierarchical structure by shifting the Swin windows to merge information between adjacent Swin windows. However, because ViTST employs a fixed per-feature normalization scheme, applying it directly to coordinate data that exhibit significant positional variability over time can obscure subtle variations in movement trajectories. Additionally, data that belong to different absolute coordinate systems but share the same local coordinate system may not be recognized as the same class. Moreover, when features correspond to observed coordinate data—which can vary widely over time—their line graph representations struggle to convey the differing scales of multiple feature axes within a single image. ViTST, which does not incorporate dynamic range embedding, therefore fails to capture this variability.

## 3 Proposed Method

Figure 2 illustrates the overall flow of the proposed training method, divided into three stages. As shown on the left, the SKETCH (Skeleton Kinematics Extraction Through grapH) module transforms raw four-dimensional skeleton data (time, $x$, $y$, and $z$) into a visual representation—enabling high-level feature extraction from the raw data without prior topology knowledge—then applies our novel Dynamic Range Embedding (DRE) to encode axis-wise relative range information, which is subsequently added via element-wise broadcasting to the corresponding partition of the image. Here, $F_t$ represents the frame corresponding to the skeleton information at time $t$, and a window $W_t$ is defined by grouping $N_F$ frames (where $N_F$ denotes the number of frames per window), which is then converted into a single graph image. In the middle stage, the graph image generated by SKETCH is used as input to the backbone, where each image is divided into patches; relationships between patches are captured via self-attention within Swin windows, and subsequently, 'patch merging' [27]

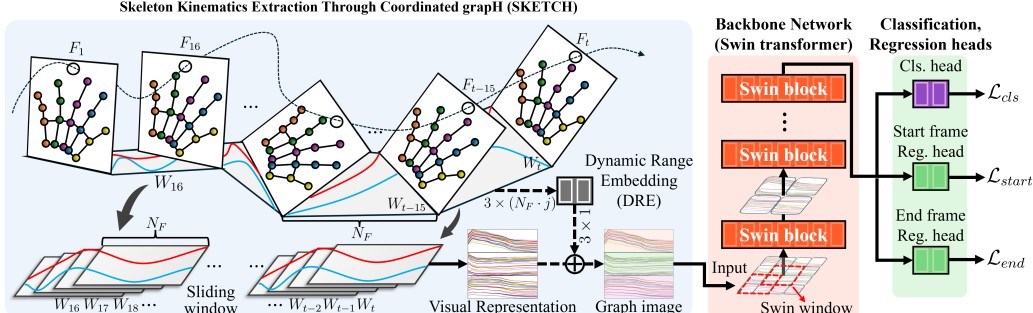

Figure 2: Overall architecture of the proposed method. First, the skeleton data are partitioned into windows ($W_t$). The SKETCH module then normalizes each window per-axis, generates a visual representation, and applies Dynamic Range Embedding to produce the final graph image. The transformed graph image is then passed through a backbone, followed by two separate heads: one for gesture classification and another for start and end frame regression, enabling simultaneous gesture recognition and segment boundary detection. $\oplus$ indicates element-wise broadcasting addition.

integrates local and global information. On the right, a classification head is responsible for gesture classification per window, and two separate regression heads estimate the start and end frames of the gesture within the window. Finally, the classification head and regression head are optimized using cross entropy loss and mean squared error (MSE) loss, respectively.

## 3.1 Transforming the Online Task into a Window Problem

As shown in Fig. 11, in this study, the continuous stream of input frames in an online environment is grouped into a window ($W$), thereby converting the real-time frame-based classification task into a window-based problem. The window $W$ is represented as $W_t \in \mathbb{R}^{N_F \times j \times c}$ using time $t$. Each window, composed of $N_F$ frames, is represented as a three-dimensional array with dimensions corresponding to the numbers of frames ($N_F$), joints ($j$), and coordinates ($c$), effectively capturing the spatio-temporal characteristics of the skeleton data. Within each window, the inherent frame-level labels ($N_F$ labels) are aggregated, and the most frequent label is assigned as the window-level label, a strategy applied during both training and testing. A window may also be labeled as 'non-gesture' if non-gesture frames are dominant. In particular, when both gesture frames and non-gesture frames are mixed within a window, the gesture's start and end frames are also labeled to regress the boundaries of the gesture, where these annotations are normalized to a $[0, 1]$ range by mapping the first and last frames of the window to 0 and 1, respectively. Aggregating frame-level labels in this way enables the assignment of a reliable representative label to each window, helping to mitigate the effects of noisy or inconsistent labels—such as transient frame-level mismatches between the actual gesture transition frames and the incorrectly labeled frames. Moreover, by converting the problem into a window-based one, leveraging the independent contextual information of each window and performing normalization within each window can reduce the domain gap between training and testing data.

## 3.2 Skeleton Kinematics Extraction Through Coordinated grapH (SKETCH) Module

Figure 3 shows the graph plotting, normalization, and Dynamic Range Embedding (DRE) employed in the proposed SKETCH module. The raw data is divided into windows ($W$), and min–max normalization is performed on each coordinate axis ($x$, $y$, and $z$) within each window. Through this process, the relative patterns and subtle differences in the data are emphasized without relying on the absolute coordinates. The normalized data is then converted into a 2D image through graph plotting. Specifically, as shown on the right side of Fig. 3, the horizontal axis represents time ($t$), while the vertical axis represents each coordinate value of ($x$, $y$, and $z$), and three subplots are vertically stacked to form a single graph image. In each subplot, the coordinates of $j$ joints are displayed as discrete points, which are connected by lines to represent continuous movement. To preserve cross-axis relative scales that may be lost in this transformation, we introduce our novel DRE to encode axis-wise relative range information. Specifically, for each axis, coordinate values across

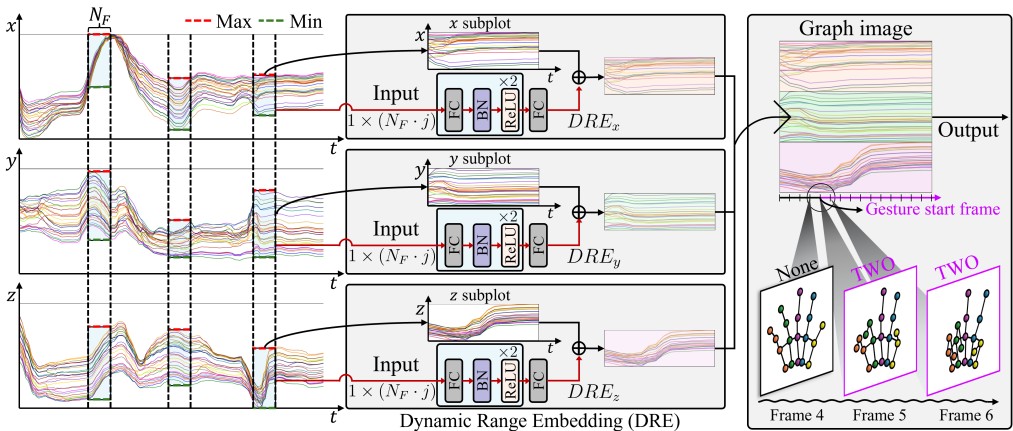

Figure 3: Process of the SKETCH module. Raw skeleton sequences are windowed and min–max normalized per coordinate axis ($x$, $y$, and $z$). In each subplot, 26 lines corresponding to the joint coordinates are displayed. The dynamic range embedding module shares weights across all coordinate axes. Frames 4, 5, and 6 show raw skeleton data aligned with the time-axis points of the graph image.

$N_F$ frames and $j$ joints undergo $Z$-normalization [35] and are then reshaped into a vector of length $\in \mathbb{R}^{1 \times (N_F \cdot j)}$. This vector is processed by two successive fully connected (FC) layers—each followed by BatchNorm (BN) and ReLU activation—and finally by a third FC layer to produce a single scalar DRE value per axis, denoted as $DRE_x$, $DRE_y$, and $DRE_z$. These scalar values, which compactly represent the movement range or variability of each axis, are broadcast [34] to all pixel channels of the corresponding axis subplot and added to the visual representation, as illustrated in Fig. 3. The entire process is termed Skeleton Kinematics Extraction Through Coordinated grapH (SKETCH).

Unlike ViTST, as described in Section 2.3, the proposed SKETCH module represents the interactions among the $j$ joint lines within a single subplot while assigning separate subplots for each coordinate axis ($x$, $y$, and $z$), as shown in Fig. 3. This joint-wise co-location preserves the relative spatial configuration between joints, enabling the model to directly perceive their spatial dependencies. Moreover, by vertically stacking the subplots, temporal information is aligned along the horizontal axis, providing a coherent temporal flow that effectively preserves the continuity of motion, thereby aiding the model in identifying precise gesture boundaries such as the start and end frames. In contrast, ViTST's feature-wise isolated line graphs fragment the spatial context and obscure inter-joint dependencies, making it difficult for the model to capture spatial relationships and temporal continuity simultaneously (further details are provided in Appendix E.1). Furthermore, each joint is consistently assigned a fixed color, ensuring consistent joint representation. Unlike conventional methods that rely either on hand-crafted features or on dataset- and sensor-specific topology definitions, our approach represents the time, three-dimensional space, and joints of the raw data in a single graph. This graph image implicitly incorporates features typically extracted through rule-based methods while enabling the model to autonomously learn richer representations from the raw data, and it is inherently topology-agnostic, ensuring robust generalization across diverse skeleton structures. In this way, by representing raw data on a window basis and performing normalization for each coordinate within the window, the distribution differences (domain gap) between the training and testing data can be mitigated. Through the SKETCH module, converting four-dimensional data into two-dimensional images enables the effective utilization of powerful pre-trained classification models.

## 3.3 Backbone Models

In this study, we adopt Vision Transformer (ViT) or Swin Transformer as backbone models, which tokenize input into patches and utilize Transformer architectures for joint gesture classification and boundary localization [11, 27]. Swin Transformer further captures both local and global dependencies via hierarchical patch merging and Swin window shifting, enabling the model to learn intra-subplot relationships (among joints sharing the same coordinate axis) and inter-subplot relationships (across

different coordinates and time steps). To support this, each backbone is equipped with a classification head and two regression heads—implemented as single linear layers—for predicting gesture types and temporal boundaries between gesture and non-gesture segments, respectively.

## 3.4 Loss Function

The model employs a combined loss function consisting of classification and regression terms. The classification loss ($\mathcal{L}_{cls}$) uses cross entropy as in [8] to classify the type of gesture, while the regression losses ($\mathcal{L}_{start}$, $\mathcal{L}_{end}$) use mean squared error (MSE) as in [8] to predict normalized start and end frame positions within each window. This formulation enables the model to simultaneously optimize gesture classification and boundary prediction. The total loss is defined as:

$$\mathcal{L} = \lambda_1 \mathcal{L}_{cls} + \lambda_2 \mathcal{L}_{start} + \lambda_3 \mathcal{L}_{end}. \tag{1}$$

## 3.5 Online Inference Process

In the inference phase, the output from the regression head is not used; only the results from the classification head are utilized. Continuous window-based predictions are post-processed following Algorithm 1 to classify frames into a certain gesture. Briefly, the input frames are grouped into windows of fixed length ($N_F$), and for each window, class prediction value ($\hat{y}$) is obtained using the model ($f$). These predictions are then accumulated in a queue structure of fixed size ($Q$), and $g$ is determined by applying a majority voting method to the class prediction values stored in the queue, enabling noise-robust frame-wise classification. However, waiting for the final detection after collecting $Q$ prediction values introduces a certain delay. Therefore, an appropriate value for $Q$ must be applied, taking into account both noise reduction and delay.

## 4 Experiments

All experiments were conducted using PyTorch [34] on a single NVIDIA RTX A6000 or NVIDIA A100 GPU, and the model was trained and tested using the SHREC'19 and SHREC'22 datasets (details provided in Appendix B). Appendix E reports the impact of various ablation versions of the SKETCH module, loss configurations, and related factors, etc. Training strategies and implementation details are provided in Appendix G.

### 4.1 SHREC'19 Results

Table 1: Comparison of benchmark and proposed methods on SHREC'19. DR and FP denote detection rate and false positive, respectively. Values in parentheses indicate backbone configurations, where backbone type: V = ViT, S = Swin; model size: L = Large, B = Base, S = Small; patch size; Swin window size (for Swin only); and image resolution. JCD, FD, Adj, and Ref indicate the use of joint collection distances, frame difference vectors, adjacency matrices, and reference-joint difference vectors features, respectively. Plot denotes the use of graph image as input. 'Aug.' indicates the use of additional data augmentation strategies, which are detailed in Appendix G.

| Method | DR↑ | FP↓ | Time (s) | FPS | JCD | FD | Adj | Ref | Plot |
|---|---|---|---|---|---|---|---|---|---|
| PSUMNet [44] | 0.64 | 0.22 | 0.0250 | 40 | ✓ | ✓ | | | |
| MS-G3D [28] | 0.69 | 0.25 | 0.0303 | 33 | ✓ | | | | |
| SeS-GCN [38] | 0.75 | 0.12 | 0.0020 | 500 | | | ✓ | | |
| SW 3-cent [3] | 0.76 | 0.19 | 0.0030 | 333 | | ✓ | | ✓ | |
| DS-GCN [51] | 0.80 | 0.05 | - | - | | | ✓ | ✓ | |
| DSTA [41] | 0.81 | 0.08 | 0.0088 | 114 | ✓ | | | | |
| DG-STA [5] | 0.81 | 0.07 | 0.0042 | 238 | | | ✓ | | |
| DDNet [54] | 0.82 | 0.10 | 0.0022 | 455 | ✓ | ✓ | | | |
| DDNet [54] | 0.82 | 0.10 | 0.0022 | 455 | ✓ | ✓ | | | |
| BlockGCN [60] | 0.83 | 0.04 | - | - | | ✓ | ✓ | ✓ | |
| uDeepGRU [4] | 0.85 | 0.10 | 0.0030 | 333 | | ✓ | | ✓ | |
| ProtoGCN [24] | 0.86 | 0.05 | 0.0334 | 30 | | | ✓ | ✓ | |
| OO-dMVMT [8] | 0.88 | 0.05 | 0.0058 | 172 | ✓ | ✓ | | | |
| SKETCH (V-L-16-384) | 0.90 | 0.03 | 0.0176 | 57 | | | | | ✓ |
| SKETCH (S-S-4-7-224) | 0.88 | 0.04 | 0.0039 | 256 | | | | | ✓ |
| SKETCH (S-B-4-12-384) | 0.91 | 0.03 | 0.0091 | 110 | | | | | ✓ |
| SKETCH (S-L-4-12-384) | 0.92 | 0.02 | 0.0142 | 70 | | | | | ✓ |
| SKETCH (S-L-4-12-384) + Aug. | 0.93 | 0.01 | 0.0142 | 70 | | | | | ✓ |

As presented in Tab. 1 and illustrated in Fig. 12, the proposed SKETCH outperforms previous SOTA methods in gesture-wise detection accuracy across diverse categories, while also ensuring real-time operation with a throughput of at least 30 frames per second (FPS). This performance is particularly noteworthy given the challenging evaluation protocol of the SHREC'19 dataset [4], which involves a clear domain shift, as the test subjects were entirely different from those used in training. Despite this domain discrepancy, the SKETCH module exhibits robust generalization ability, suggesting its effectiveness in mitigating the domain gap—more so than existing methods.

## 4.2 SHREC'22 Results

Table 2: Recognition performance on the SHREC'22. JI denotes Jaccard index. Configurations follow the same notation as described in Tab. 1.

| Method | DR↑ | FP↓ | JI↑ | Delay (fr.) | Time (s) | JCD | FD | Adj | Plot |
|---|---|---|---|---|---|---|---|---|---|
| DeepGRU [30] | 0.26 | 0.25 | 0.21 | 8.0 | 0.0031 | | | | |
| DG-STA [5] | 0.51 | 0.32 | 0.40 | 8.0 | 0.0042 | | | | ✓ |
| SeS-GCN [38] | 0.60 | 0.16 | 0.53 | 8.0 | 0.0018 | | | | ✓ |
| PSUMNet [44] | 0.62 | 0.24 | 0.52 | 8.0 | 0.0244 | ✓ | ✓ | | |
| MS-G3D [28] | 0.68 | 0.21 | 0.57 | 8.0 | 0.0293 | ✓ | | ✓ | |
| Stronger [12] | 0.72 | 0.34 | 0.59 | 14.8 | 0.1000 | ✓ | ✓ | | |
| DSTA [41] | 0.73 | 0.24 | 0.61 | 8.0 | 0.0092 | ✓ | | ✓ | |
| 2ST-GCN+5F [12] | 0.74 | 0.23 | 0.61 | 13.3 | 0.0021 | ✓ | ✓ | | |
| TN-FSM+JD [12] | 0.77 | 0.23 | 0.63 | 10.0 | 0.0046 | ✓ | | ✓ | |
| Causal TCN [12] | 0.80 | 0.29 | 0.68 | 19.0 | 0.0280 | | | | ✓ |
| DDNet [54] | 0.88 | 0.16 | 0.78 | 8.0 | 0.0022 | ✓ | ✓ | | |
| OO-dMVMT [8] | 0.92 | 0.09 | 0.85 | 8.0 | 0.0041 | ✓ | ✓ | | |
| SKETCH (S-B-4-12-384) | 0.91 | 0.06 | 0.86 | 8.0 | 0.0097 | | | | ✓ |
| SKETCH (S-L-4-12-384) | 0.92 | 0.07 | 0.87 | 8.0 | 0.0124 | | | | ✓ |
| SKETCH (S-L-4-12-384) + Aug. | 0.95 | 0.06 | 0.91 | 8.0 | 0.0124 | | | | ✓ |

Table 2 presents the classification results of our method on the SHREC'22 dataset [12], demonstrating competitive performance against SOTA approaches. Unlike conventional methods that rely on hand-crafted features, our approach leverages SKETCH-based visual plots for a more interpretable representation. As illustrated in Figs. 7, 13, and 14, our method effectively distinguishes visually similar dynamic gestures such as Circle, V, and Cross—tasks where hand-crafted feature-based methods often struggle due to their limited use of raw skeletal data.

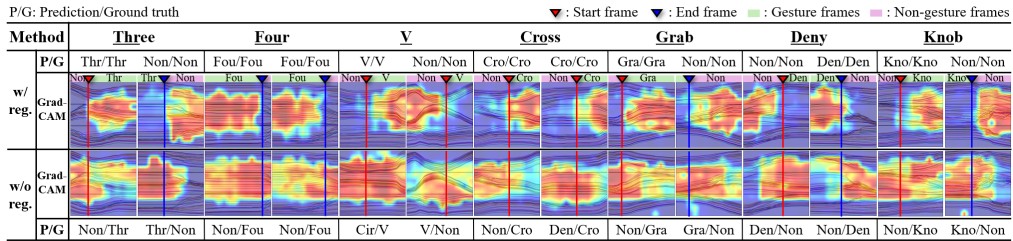

Figure 4: Grad-CAM visualizations illustrating how boundary regression loss enhances the model's ability to localize gesture boundaries. Each Grad-CAM image corresponds to a single input window. The red and blue lines represent the start and end frames of the gestures within each window, respectively. For simplicity, abbreviated gesture names (e.g. Three → Thr) are used in the figure.

Figure 4 shows the results of Grad-CAM [40] to visualize the impact of the regression losses ($\mathcal{L}_{start}$ and $\mathcal{L}_{end}$) on the classification process. By detecting the start and end frames of a gesture, the model learns localized boundaries rather than performing global classification. This enables refined discrimination even in graphs where gestures and non-gestures are mixed, making it difficult to judge a single classification based solely on the overall gesture. In contrast, models trained without regression loss exhibit a tendency to attend to broader temporal regions, overlooking the actual boundaries between different consecutive gestures. This often leads to ambiguities in temporal localization and increases the likelihood of misclassifying gestures, particularly when gesture transitions are subtle or occur in quick succession. Such boundary-aware modeling allows for more adaptive and precise

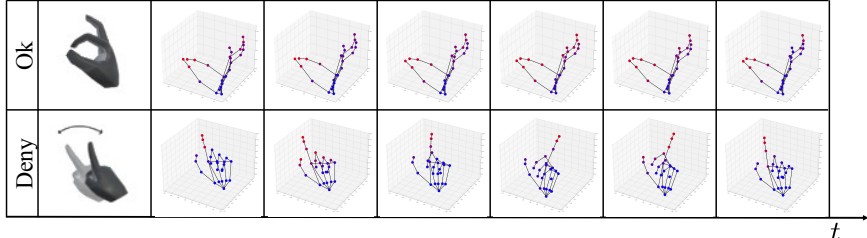

Figure 5: Visualization of joint-level attention for gestures in the SHREC'22 dataset. The color intensity indicates the magnitude of attention assigned by the model to each joint, with attention increasing progressively from blue to red (see Fig. 15 for extended visualizations).

predictions compared to classification-only approaches. In addition, it opens up new perspectives on the interpretability of features, thereby enhancing the reliability of gesture recognition.

Humans distinguish hand gestures by focusing on key joints or finger movements that uniquely define each gesture. To examine whether the model attends to gestures in a similar way, we applied Grad-CAM to the SKETCH-based graph images and mapped the resulting attention scores back to the 3D skeleton coordinates for quantitative evaluation. As illustrated in Figs. 5 and 15, attention maps reveal that the model focuses on task-relevant joints in a human-like manner, suggesting that it captures high-level semantic representations rather than overfitting to geometric biases. This supports the interpretability and intuitiveness of our SKETCH-based graph transformation. For further details on the attention mapping process, qualitative interpretation, see the description in Fig. 15. In addition, gesture-specific attention patterns and their semantic alignment are thoroughly analyzed in Fig. 8.

## 5 Conclusion

In this study, we proposed the Skeleton Kinematics Extraction Through Coordinated grapH (SKETCH) module to overcome the limitations of conventional hand-crafted feature-based approaches in skeleton-based hand gesture recognition. The proposed method converted four-dimensional (time, $x$, $y$, and $z$) skeleton raw data into simple and intuitive visual graphs, with Dynamic Range Embedding (DRE) further preserving axis-wise relative scale during this transformation, demonstrating that it could fully leverage the rich information that is easily overlooked by conventional methods. In particular, while existing approaches provided non-intuitive and uninterpretable representations using a sequence of data, the SKETCH module opened up a new perspective in hand gesture recognition through an interpretable data representation. Moreover, by applying an online, independent min–max normalization technique, it effectively mitigated the domain gap caused by changes in sensor viewpoints and individual body proportion differences, thereby achieving consistent performance across various test environments. Through these designs, the framework becomes topology-agnostic, ensuring robust generalization across diverse skeleton structures. Experimental results showed that the proposed SKETCH method outperformed existing approaches in terms of high detection and classification performance on major benchmarks such as SHREC'19 and SHREC'22. These results suggested that skeleton-based hand gesture recognition can play a key role in realizing intuitive human–computer interaction in fields such as intelligent robotics and mixed reality systems, and our approach demonstrated the potential for extension to various future applications.

### 5.1 Limitations and Future Work

Although we demonstrated that high-level features can be extracted directly from raw skeleton data with our novel representation, the current architecture still makes each decision independently and therefore ignores information from preceding windows. In future work, we plan to overcome this limitation by moving beyond fixed-length chunks to an adaptive scheme that captures longer-range temporal context, ultimately enabling a unified architecture suitable for both online and offline 4D action recognition tasks. Moreover, while our evaluation focused on skeleton-based hand gesture recognition, the topology-agnostic design of SKETCH suggests its potential applicability to broader domains such as skeleton-based human action recognition, which we leave as promising directions for future research.

# 6 Acknowledgements

This research was supported by the National Research Foundation (NRF) funded by the Korean government (MSIT) under Grant RS2023-00208763, and RS-2025-02653611 (Intelligent Semiconductor Technology Development Program).

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

# Appendix

# Table of Contents

# A Overcoming Geometric Shortcut and Coordinate Bias via SKETCH

In prior 3D-based studies, there have been continued efforts to encourage models to autonomously learn rich representations directly from unprocessed raw point data, as discussed in [56]. In particular, coordinate-based 3D inputs such as skeletal joints and point clouds are commonly processed by feeding normalized coordinate values directly into the model, without any intermediate representational transformation. However, this approach often leads to an excessive reliance on the absolute coordinates of the training data, causing the model to overfit to low-level geometric cues [6, 50, 52] rather than learning semantically meaningful representations.

This phenomenon has also been clearly identified in recent work such as Sonata [50], which highlights the geometric shortcut as a key limitation in 3D self-supervised learning [50, 52, 56]. The geometric shortcut refers to the model's tendency to overly rely on low-level spatial hints such as surface normal vectors, point heights, and relative positions, rather than learning meaningful semantic features. This issue is particularly pronounced in 3D point cloud data, which is inherently sparse and non-grid in structure, making it more likely for coordinate values to be misinterpreted as semantic cues. For example, in environments where objects like the floor, desk, and ceiling can be distinguished solely based on their $y$-coordinate (height), models may memorize these positional cues instead of learning abstract, high-level representations. As a result, the learned features tend to collapse into simple geometric patterns, ultimately degrading the model's generalization ability.

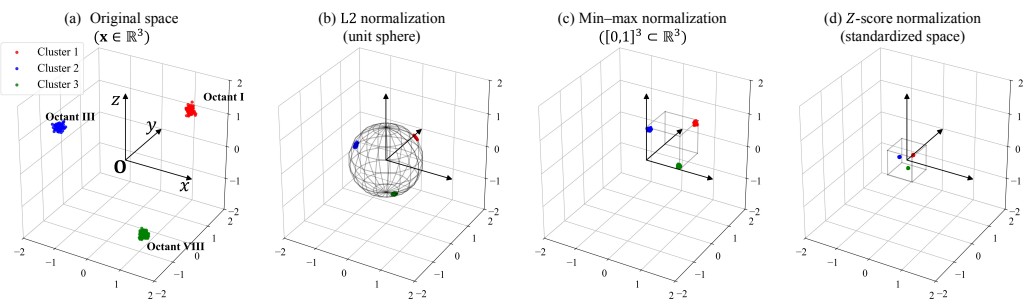

Figure 6: Comparison of raw and normalized 3D spaces using synthetic point clusters, demonstrating that the relative spatial relationships between clusters persist across different normalization methods. Black boundaries indicate each normalization space: (a) original space with three color-coded clusters in Octants I, III, and VIII; (b) L2 normalization on the unit sphere; (c) min–max normalization in the $[0, 1]^3$ cube; (d) $Z$-score normalization in a standardized $\pm 3\sigma$ cube.

Furthermore, Fig. 6 illustrates that simple normalization alone is insufficient to fundamentally eliminate coordinate bias [56]. The figure shows red, blue, and green point clusters located in Octant I ($x > 0, y > 0, z > 0$), Octant III ($x < 0, y < 0, z > 0$), and Octant VIII ($x > 0, y < 0, z < 0$) of the original three-dimensional space, respectively, and demonstrates that these clusters retain their relative spatial configuration even after normalization. Whether L2 normalization [39], min–max scaling, or $Z$-score normalization [35] is applied, the directional and positional relationships between clusters remain preserved, thereby allowing the model to continue learning features based on coordinate-dependent spatial cues. Therefore, applying normalization to the raw data still fails to resolve the geometric shortcut problem.

For example, in the case of an arbitrary spatial movement over time (e.g., Octant I → III → VIII), the same dynamic gesture may be observed as different coordinate trajectories (e.g., Octant III → I → VIII) depending on the viewpoint. Although the underlying motion remains identical, the resulting coordinate trajectory differs due to changes in the observation angle. In other words, the same gesture can be represented by entirely different coordinate sequences depending on the viewpoint. Even after normalization, the relative arrangement and orientation of the points are preserved, allowing the model to learn low-level representations based on such spatial cues. As a result, normalization alone is insufficient to eliminate viewpoint-induced variation, and it fails to prevent the model from overfitting to geometric shortcuts based on specific positions or directions. A similar phenomenon can also be observed in Static Gestures (SG). Even when SG are performed within a single octant, the orientation

of the hand may vary depending on the sensor's viewpoint. As a result, the same gesture may exhibit different inter-joint spatial relationships—for instance, the $z$-coordinate of the index finger may appear greater or smaller than that of the thumb. Such variation arises not only from changes in viewpoint but also from individual differences in body proportions. This indicates that simple normalization is insufficient for achieving viewpoint-agnostic or subject-invariant representation learning.

Such issues have also been observed in prior studies on hand gesture recognition. For example, models like deepGRU [30], which rely (solely) on raw coordinate inputs, have exhibited significantly degraded performance due to representation collapse.

To address these challenges, this study proposes a plotting-based intermediate medium (SKETCH) that goes beyond simple coordinate normalization and transforms normalized coordinates into representations that alleviate sensor viewpoint and subject-specific variations. Specifically, raw point or joint data are converted into graph images, which are then processed by pretrained image backbones (e.g., ViT or Swin Transformer) to effectively extract high-level semantic features in the image domain. This approach enables more effective utilization of the representational potential of raw 3D data, addressing the geometric shortcut problem inherent in direct coordinate input while providing a robust pathway for expressive representation learning.

# B   Dataset Description

The datasets used for training and evaluation were both online datasets: SHREC'2019 [4] and SHREC'2022 [12]. The SHREC 2019 dataset captures five distinct hand gestures—Cross (X), Circle (O), V-mark (V), Caret (^), and Square ([ ])—performed by 13 users using a LeapMotion sensor at 0.05-second intervals. Each frame records 3D positions ($x$, $y$, and $z$) and quaternions ($x$, $y$, $z$, and $w$) from 16 hand joints, though only the positional data is used in our approach for gesture classification and gesture boundary prediction. Data from four users are used for training, and data from the remaining nine users are used for testing. One sequence missing annotation was excluded from the performance evaluation.

The SHREC'2022 dataset comprises 144 training sequences and 144 testing sequences captured with the Hololens2 device [12], with each sequence consisting of 3D coordinates (x, y, and z) for 26 joints across frames of varying lengths. It contains 16 types of gestures, as depicted in the Fig. 7, which are further classified into four categories: Static (SG: One, Two, Three, Four, Ok, and Menu), Dynamic (DG: Left, Right, Circle, V, and Cross), Fine-grained Dynamic (FG-DG: Grab and Pinch), and Dynamic-periodic gestures (D-PG: Deny, Wave, and Knob).

The SHREC'19 online-gesture dataset is publicly available for non-commercial academic use (no explicit license is specified by the organizers), whereas the SHREC'22 dataset is released under the Creative Commons Attribution–NonCommercial–NoDerivatives 4.0 International (CC BY-NC-ND 4.0) license.

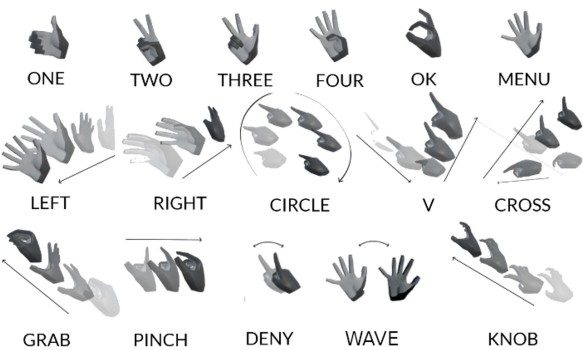

Figure 7: Illustration of the 16 gesture classes in the SHREC'2022 dataset [12], categorized into Static, Dynamic, Fine-grained Dynamic, and Dynamic-periodic gestures.

# C Evaluation Metrics

To quantitatively evaluate the performance of gesture recognition, we adopted four evaluation metrics: detection rate (DR), false positive (FP) rate, Jaccard index (JI), and delay. These metrics were computed for each gesture class and averaged across all classes to obtain the final results. The evaluation protocol strictly follows the official metrics defined in the SHREC'22 [12] online gesture recognition track. A predicted gesture segment was considered a correct detection if it satisfied the following conditions: The predicted label matches the ground-truth label. The predicted segment length does not exceed twice the ground-truth segment length. The temporal (frame) overlap ratio between the predicted segment $[f_p^s, f_p^e]$ and the ground-truth segment $[f_g^s, f_g^e]$ is greater than or equal to 0.5:

**Detection criterion** A predicted gesture segment is considered a correct detection if it satisfies all of the following conditions:

1. The predicted label matches the ground-truth label.

2. The predicted segment length does not exceed twice the ground-truth segment length.

3. The temporal (frame) overlap ratio between the predicted segment $[f_p^s, f_p^e]$ and the ground-truth segment $[f_g^s, f_g^e]$ is at least 0.5:

$$\text{overlap} = \frac{\min(f_g^e, f_p^e) - \max(f_g^s, f_p^s)}{f_g^e - f_g^s} \ .$$

**Evaluation metrics** Based on the above matching rule, we define the following metrics:

- **Detection rate (DR):**

$$\text{DR} = \frac{\text{TP}}{\text{GT}},$$

  where TP is the number of correctly detected gesture segments, and GT is the total number of ground-truth gesture segments.

- **False positive rate (FP rate):**

$$\text{FP} = \frac{\text{FP}_{\text{count}}}{\text{GT}},$$

  where $\text{FP}_{\text{count}}$ is the number of unmatched predicted gestures.

- **Jaccard index (JI):**

$$\text{JI} = \frac{\text{TP}}{\text{TP} + \text{FN} + \text{FP}},$$

  where FN is the number of missed gesture segments.

- **Delay:**

$$\text{Delay} = \begin{cases} \left\lfloor \dfrac{Q}{2} \right\rfloor - 1, & \text{if } Q \text{ is even,} \\ \left\lfloor \dfrac{Q}{2} \right\rfloor, & \text{if } Q \text{ is odd.} \end{cases}$$

  where $Q$ is the queue size in Algorithm 1.

All metrics were calculated for each gesture class and the overall performance was computed as the mean value across all gesture classes. This evaluation protocol enables a comprehensive assessment of detection accuracy, robustness to false positives, segmentation quality, and temporal prediction latency.

# D  Gesture-Specific Attention Patterns and Interpretability Analysis

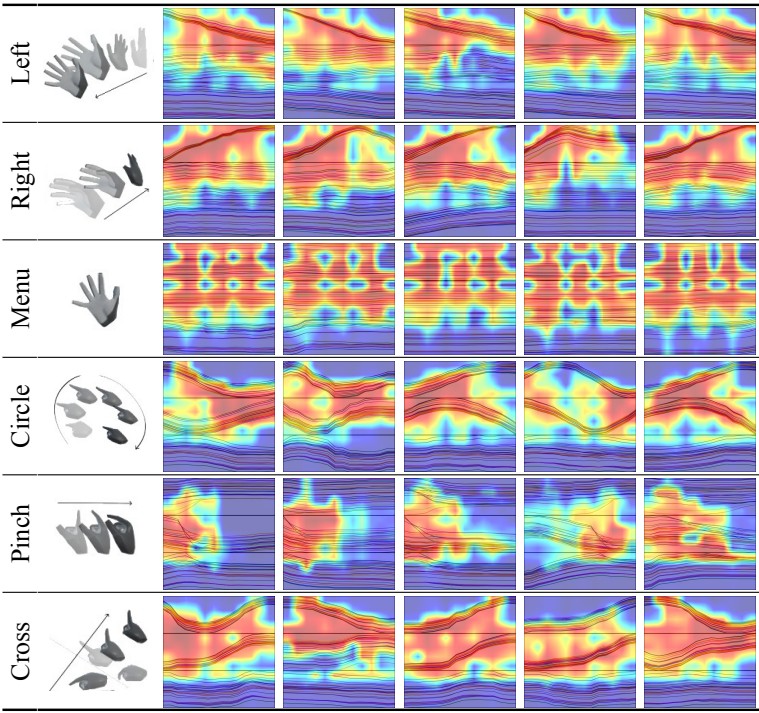

Figure 8: Visualization of model's attention during gesture prediction for the Left, Right, Menu, Circle, Pinch, and Cross gestures using Grad-CAM [40]. All examples were extracted from different sequences.

Figure 8 illustrates Grad-CAM [40] result on the hand gesture recognition process through the SKETCH. It demonstrates that the unique motion patterns of each gesture align with the regions that the model focuses on, thereby showcasing the interpretability of our proposed method—enabled by its use of raw data—in contrast to conventional, uninterpretable hand-crafted features. In this manner, the transformation process using the SKETCH technique reflects the intrinsic characteristics of each gesture and confirms that it forms an interpretable structure that consistently aligns with the regions on which the model focuses.

For instance, the Left and Right gestures are characterized by the hand's overall movement toward the left or right direction while the hand shape itself remains fixed; during the SKETCH transformation, a linearly increasing or decreasing graph is formed in the $x$-coordinate subplot. Accordingly, the model's attention is observed to align closely with these linear patterns in the graph. The Menu gesture, defined by a static open-handed posture, exhibits a pattern of widely spaced lines distributed across the entire image, suggesting that capturing the overall structural pattern plays a more critical role in prediction than relying solely on localized information—an observation further supported by the model's behavior. The Circle gesture involves circular movement of the hand without altering its shape. This results in waveform patterns appearing on both the $x$- and $y$-coordinate subplots after the SKETCH transformation. Thus, it is confirmed that the model focuses on the regions where these waveforms persist. The Pinch gesture is distinguished by the convergence of the thumb and index finger coordinates, and the SKETCH transformation results in a pattern where the lines in the $x$ and $y$ subplots gradually widen. The model's emphasis on the widening region indicates that this is a critical element in distinguishing the Pinch gesture.

# E   Ablation Studies

## E.1   Ablation Study: Plotting Methods

Table 3: Quantitative comparison on the SHREC'22 dataset of gesture classification error rates (%) across four gesture categories—Static Gestures (SG), Dynamic Gestures (DG), Fine-grained Dynamic Gestures (FG-DG), and Dynamic-periodic Gestures (D-PG)—under different ablation settings of the proposed SKETCH module, including removal of color, ViTST-style joint coordinate arrangement, and addition of markers. The values in parentheses "( )" indicate the error rate increase compared to the standard SKETCH module.

| Gestures | SKETCH | (a) Black | (b) ViTST | (c) Marker |
|---|---|---|---|---|
| SG | 6.02 | 15.74 (+9.72) | 29.63 (+23.61) | 8.80 (+2.78) |
| DG | 5.00 | 8.89 (+3.89) | 7.78 (+2.78) | 7.78 (+2.78) |
| FG–DG | 12.50 | 15.28 (+2.78) | 30.56 (+18.06) | 15.28 (+2.78) |
| D–PG | 13.89 | 20.37 (+6.48) | 42.59 (+28.70) | 19.44 (+5.55) |

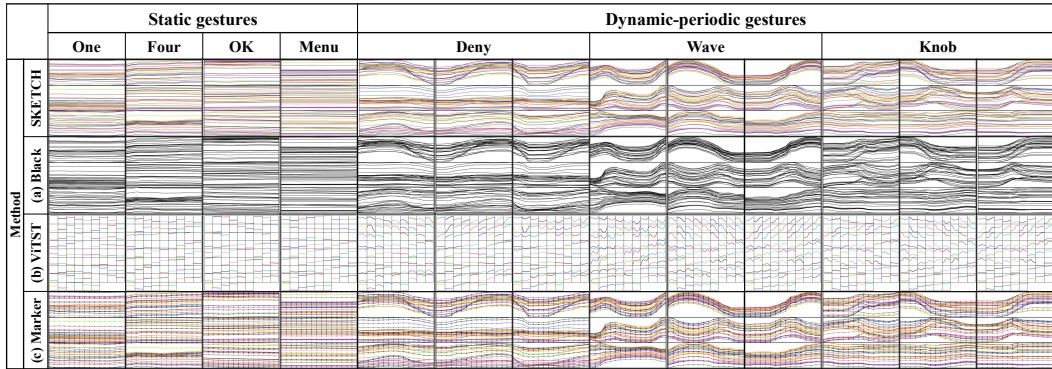

Figure 9: Visualization of graph image representations generated by different ablation versions of the SKETCH module for both static gestures (left) and dynamic-periodic gestures (right). Each row corresponds to a different visualization method: the standard SKETCH design (SKETCH), a version without color differentiation (Black), a version adopting ViTST-style joint arrangement (ViTST), and a version with added markers (Marker). For dynamic-periodic gestures three temporally consecutive and non-overlapping images are shown for each class to better capture and visualize the underlying periodic patterns. Each graph image corresponds to a single input window.

Table 3 presents a quantitative comparison of error rates across four gesture categories (SG, DG, FG-DG, D-PG) under various ablation settings of the proposed SKETCH module. The experiments were conducted under three ablation configurations: (a) removing color differentiation by rendering all joints in the same black color, (b) replacing the joint coordinate arrangement of SKETCH with the ViTST [22] layout (i.e., representing each joint as an individual line graph), and (c) adding markers at each observation point to emphasize the observed data. It should be noted that in setting (b), the full plotting method of ViTST, as described in Section 2.3, was not adopted. Instead, the joint coordinates were rearranged into ViTST-style individual line graphs by treating each joint coordinate as an independent feature, while still applying our proposed normalization method. The group-wise gesture classification error rate (Group Error Rate) is computed based on the prediction performance of all gesture classes belonging to the group (Static Gestures (SG), Dynamic Gestures (DG), Fine-grained Dynamic Gestures (FG-DG), and Dynamic-periodic Gestures (D-PG)) and is defined as follows:

$$\text{GroupErrorRate}_G = 1 - \frac{\sum_{i \in G} \text{TP}_i}{\sum_{i \in G} \text{Seg}_i},$$

Here, $G$ denotes a specific gesture group (e.g., FG–DG), and $i$ refers to the $i$-th gesture class (e.g., Grab) belonging to group $G$. $\text{TP}_i$ represents the number of correctly predicted segments (true positives) for class $i$, while $\text{Seg}_i$ denotes the total number of ground-truth segments for that class. In other words, the group error rate is calculated based on the overall prediction accuracy aggregated across all gesture classes within the group.

Based on the analysis, ablation setting (a) resulted in the largest error-rate increase for SG, followed by D–PG, while setting (b) caused the greatest increase for D–PG, followed by SG. These results are closely related to the inherent characteristics of SG and D–PG. Gesture classes belonging to SG involve maintaining a specific hand shape over a period of time, with little to no noticeable movement over time. As shown in the SKETCH visualizations in Fig. 9, the graph images of SG classes (e.g., One, Four, OK, and Menu) produced by the SKETCH module display joint coordinates as horizontally aligned parallel lines. This visual pattern reflects the static nature of these gestures, with minimal variation along the time axis.

Therefore, the distinction between different SG classes relies not on temporal dynamics, but rather on the relative spatial relationships between joint coordinates captured in the graph image. In this context, the SKETCH module's strategy of consistently assigning a unique color to each joint plays a critical role in differentiating the joints. The relative spatial configuration—such as a particular colored joint appearing higher than others within the same coordinate subplot, or two joints being positioned closely or far apart—serves as a key visual cue for identifying SG gesture classes. In this context, ablated version (a), which renders all joints in the same color (black), eliminates the ability to distinguish between joint coordinates, ultimately hindering the model's ability to effectively learn the unique spatial arrangement (permutation) of joints characteristic of each SG class. In contrast, ablated version (b) modifies the joint coordinate layout of SKETCH to follow the ViTST approach, where each joint is represented as an individual line graph. This deviates from the standard SKETCH format, in which multiple joint coordinates sharing the same coordinate axis are visualized together within a single subplot. As a result, it becomes more difficult to capture the relative positions of joints within a localized region of the image. Instead of directly perceiving spatial relationships among joint coordinates, the model is forced to infer them indirectly based on the positional patterns within each individual line graph. Consequently, compared to the original method that visualizes multiple joints on the same coordinate axis within a single subplot, this configuration introduces limitations in effectively recognizing the spatial relationships among joints.

A defining characteristic of gestures in the D-PG category (e.g., Deny, Wave, and Knob) is their distinct periodicity, as clearly visualized in Fig. 9 using the SKETCH module. Effectively recognizing such periodic gestures requires that the shape and repetition of each cycle along the time axis be visualized with sufficiently high resolution. However, in ablated version (b), the joint coordinate layout is modified according to the ViTST format, where each joint is represented as an individual line graph arranged in a grid within a single graph image. This configuration leads to physical compression of the time axis (in terms of pixel space), as each line graph with the same time length is constrained within the limited image area. This becomes especially problematic for D-PG gestures, as subtle temporal variations and periodic patterns along the time axis are crucial for accurate gesture discrimination. If these patterns are visualized at low temporal resolution, their fine-grained structures may become distorted or lost, ultimately hindering the model's ability to correctly recognize D-PG gestures.

While (c) caused a slight performance drop for all four categories due to markers subtly distorting the fine shape of the lines and the relationships between joints. These results highlight the effectiveness of SKETCH in preserving critical spatio-temporal and structural cues through carefully designed visual representations, enabling more robust gesture recognition across diverse gesture categories.

## E.2 Ablation Study: Impact of Regression Loss, Loss Weights, and Window Length

Table 4: Quantitative comparison on the SHREC'22 dataset of gesture recognition performance with and without regression loss for gesture boundary localization.

| Method | DR↑ | FP↓ | JI↑ |
|---|---|---|---|
| w/ regression | 0.92 | 0.07 | 0.87 |
| w/o regression | 0.88 | 0.09 | 0.82 |

Table 5: Influence of loss function weights ($\lambda_1$, $\lambda_2$, and $\lambda_3$) on overall model performance on the SHREC'22 dataset.

| $\lambda_1$ | $\lambda_2$ | $\lambda_3$ | DR↑ | FP↓ | JI↑ |
|---|---|---|---|---|---|
| 1.0 | 0.5 | 0.5 | 0.90 | 0.07 | 0.85 |
| 1.0 | 1.0 | 1.0 | 0.92 | 0.07 | 0.87 |
| 1.0 | 1.5 | 1.5 | 0.88 | 0.06 | 0.84 |

Table 6: Effect of window length ($N_F$) on recognition performance on the SHREC'22 dataset.

| $N_F$ | DR↑ | FP↓ | JI↑ |
|---|---|---|---|
| 8 | 0.84 | 0.10 | 0.77 |
| 12 | 0.90 | 0.10 | 0.82 |
| 16 | 0.92 | 0.07 | 0.87 |
| 20 | 0.89 | 0.08 | 0.83 |

Based on the qualitative analysis of Fig. 4 in Section 4.2, we quantitatively compared the impact of regression loss on the classification process as shown in Tab. 4. The experimental results showed that the detection rate (DR) increased by 0.04, the false positive (FP) rate decreased by 0.02, and the Jaccard index (JI) improved by 0.05, confirming that the regression loss contributes to enhancing the model's overall prediction performance across all evaluation metrics. Furthermore, the experimental results in Tab. 5 indicate that optimal performance is achieved when regression loss and classification loss are balanced.

In Tab. 6, the window-based approach is useful for online tasks as it mitigates noise and preserves spatio-temporal context. Analysis of performance across varying window sizes confirmed that an appropriate window size improves real-time classification performance.

### E.3 Ablation Study: Conventional Hand-Crafted and Proposed Features

Table 7: Quantitative comparison of plotting-based gesture recognition performance on SHREC'22 using different input features.

| Plotting features | DR↑ | FP↓ | JI↑ | Hand-crafted | Raw |
|---|---|---|---|---|---|
| JCD | 0.80 | 0.14 | 0.73 | ✓ | |
| FD | 0.87 | 0.06 | 0.83 | ✓ | |
| SKETCH (ours) | 0.92 | 0.07 | 0.87 | | ✓ |

All features were visualized using the same SKETCH-based plotting and model pipeline to ensure a fair comparison. The quantitative results are summarized in Tab. 7. While JCD and FD are traditional hand-crafted features commonly used in prior work, their performance falls short when compared to directly plotted raw joint coordinates, indicating that hand-crafted preprocessing may lead to information loss even under identical visualization and modeling setups. This result highlights a fundamental limitation of hand-crafted representations, which rely on predefined rules and may fail to capture the full complexity of skeletal motion. In contrast, by directly utilizing raw coordinate data, our approach enables the model to autonomously extract discriminative features from rich, unprocessed inputs, thereby achieving superior recognition performance that is not solely attributable to the visualization method.

### E.4 Ablation Study: Normalization Strategies

Table 8: Impact of normalization strategies on gesture recognition performance in SHREC'19.

| Target | Axis-wise | Z-norm | Min–max norm | DR↑ | FP↓ |
|---|---|---|---|---|---|
| Window | | | ✓ | 0.85 | 0.01 |
| | ✓ | ✓ | | 0.84 | 0.02 |
| Window (ours) | ✓ | | ✓ | 0.92 | 0.02 |

Axis-wise normalization, which treats $x$, $y$, and $z$ dimensions independently, consistently outperforms joint normalization across all coordinates, as it better preserves directional variations specific to each axis. The superiority of min–max over $Z$-score normalization lies in its ability to preserve the relative spatial and temporal variations of joint movements within each window. Specifically, min–max normalization maps each coordinate value to a fixed $[0, 1]$ interval based on the observed minimum and maximum values along each axis ($x$, $y$, and $z$), thereby maintaining the relative displacement between joints at a given frame (inter-joint spatial structure) as well as the variability in a joint's trajectory over time (intra-joint temporal dynamics). Therefore, as commonly practiced in prior work ViTST [22], $Z$-score normalization is often accompanied by outlier removal techniques such as interquartile range (IQR), as used in [22], to mitigate the influence of extreme values. In our experiments, for the $Z$-norm condition, outliers were removed using the IQR method, where the interquartile range was defined as $IQR = Q3 - Q1$, and values outside the range of $[Q1 - 1.5 \times IQR, Q3 + 1.5 \times IQR]$ were excluded. After removing outliers, missing joint values were reconstructed using an interpolation process to ensure continuity in the plotted line graphs (i.e. graph image). However, in skeleton-based hand gesture recognition, where the number of joints is limited and each joint trajectory may contain critical semantic cues, removing even a few outlier joint values can disrupt the temporal continuity and degrade the model's ability to recognize subtle and continuous motion patterns essential to accurate gesture interpretation. These results empirically validate our claim that window-level, axis-wise, and min–max normalization provides increased robustness to changes in viewpoint, user scale, and sensor positioning, while serving as a critical component in achieving stable recognition in diverse environments.

### E.5 Ablation Study: Dynamic Range Embedding

Table 9: Ablation study on the effects of including Dynamic Range Embedding (DRE) and alternative plotting methods on the SHREC'19 benchmark dataset.

| Method | DR↑ | FP↓ |
|---|---|---|
| (a) Black | 0.91 | 0.04 |
| (b) ViTST | 0.75 | 0.03 |
| (c) Marker | 0.9 | 0.03 |
| (d) SKETCH w/o DRE (ours) | 0.89 | 0.03 |
| (e) SKETCH w/ DRE (ours) | 0.92 | 0.02 |

Inspired by the customized positional encoding schemes [26, 42, 46, 48, 49, 55, 58, 59] developed in diverse 3D domains to capture intrinsic spatial information, we devised a novel method that embeds the unique characteristics of each coordinate axis into a visual representation. Table 9 presents an ablation study on the SHREC'19 dataset, evaluating both plotting methods and the Dynamic Range Embedding (DRE) module across five configurations: (a) a ViTST baseline applied directly to raw joint-coordinate sequences; (b) SKETCH-black, in which all joints are rendered in uniform black; (c) SKETCH-marker, attaching markers to each joint coordinate at every observation point (reference plots for variants (a)–(c) using the same method are shown in Fig. 9); (d) SKETCH without DRE; and (e) SKETCH with DRE. Variants (b) do not utilize DRE. The results indicate that converting each skeleton sequence into a visual representation and then applying the DRE module more effectively encodes axis-wise movement range than the version without DRE, enabling SKETCH to extract higher-level features.

## E.6 Ablation Study: Robustness on Pose Estimation Errors

Table 10: Impact of perturbations and missing rates on SHREC'19 gesture recognition performance.

| Perturbations | Missing Rate | DR↑ | FP↓ |
|---|---|---|---|
| - | - | 0.92 | 0.02 |
| Sequence | 30% | 0.91 | 0.02 |
| Sequence | 50% | 0.89 | 0.02 |
| Sequence | 70% | 0.85 | 0.02 |
| Sequence | 90% | 0.73 | 0.34 |
| Frame | 30% | 0.90 | 0.02 |
| Frame | 50% | 0.78 | 0.20 |
| Frame | 70% | 0.06 | 0.00 |

Table 11: Robustness on SHREC'19 under noise perturbations with varying noise rates (std.).

| Perturbations | Std. | DR↑ | FP↓ |
|---|---|---|---|
| Noise | 0.02 | 0.89 | 0.04 |
| Noise | 0.03 | 0.89 | 0.04 |
| Noise | 0.05 | 0.76 | 0.07 |

To further assess the robustness of SKETCH to pose estimation errors, we conducted additional evaluations under missing-data perturbations on SHREC'19, introduced only at test time to examine generalization under degraded pose inputs. Specifically, we considered three types of perturbations. First, missing-joint robustness was tested by simulating joint-sensor failure through random deletion of joints, and second, occlusion robustness was examined by masking joint coordinates across both temporal and spatial dimensions to simulate partial marker occlusions. The results of these two settings are summarized in Tab. 10. Finally, test noise robustness was evaluated by injecting white Gaussian noise into the training joint coordinates and subsequently testing the model on sequences corrupted with the same noise distribution, as shown in Tab. 11.

As summarized in Tabs. 10 and 11, we report detection rate (DR) and false positives (FP) under all three perturbation settings. Despite these synthetic degradations, SKETCH consistently demonstrated stable performance by capturing the spatio-temporal flow of joint kinematics, rather than relying on individual joint positions, thereby highlighting its resilience to pose estimation errors.

## E.7 Ablation Study: Regression Head and Dynamic Range Embedding

Table 12: Ablation study of our proposed modules on the SHREC'19 benchmark dataset.

| Method | Regression | DRE | DR↑ | FP↓ |
|---|---|---|---|---|
| | | | 0.84 | 0.04 |
| SKETCH | ✓ | | 0.89 | 0.03 |
| | | ✓ | 0.90 | 0.04 |
| SKETCH (ours) | ✓ | ✓ | 0.92 | 0.02 |

As detailed in Sections E.2 and E.5, this section we conducted an ablation study on the SHREC'19 dataset to evaluate the individual contributions of the regression head and Dynamic Range Embedding (DRE). We demonstrate that incorporating regression loss (see Equation 1) into the feature extraction process during training is vital for effective hand gesture recognition, while our visual representation, enhanced by DRE, effectively encodes the relative movement range and variability of joint coordinates. By enabling the model to distinguish whether a motion is ongoing or has ended, and to represent the dynamic behavior of joint coordinate trajectories, SKETCH is able to extract high-level spatiotemporal

features directly from raw data. Accordingly, Tab. 12 reports performance with and without each component: the full model—including both the regression head and DRE—consistently achieves the highest accuracy and the lowest false positive rate.

## F  Additional Visualizations

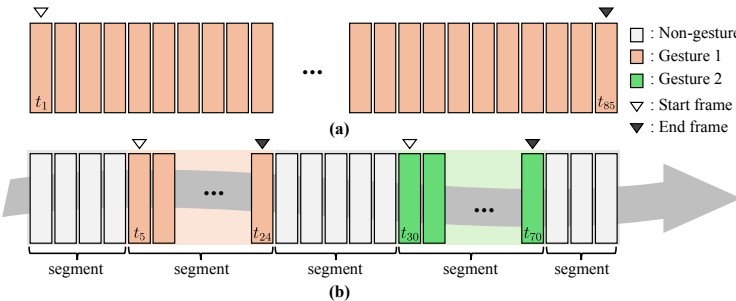

Figure 10: Comparison of offline and online datasets. (a) In the offline dataset, all frames within a sequence correspond to a single gesture. (b) In the online dataset, non-gesture segments exist, and each individual frames may belong to different gesture classes within the same sequence.

As illustrated in Fig. 10 and discussed in Sections 2.1 and 2.2, Fig. 10(a) represents an offline dataset setting, in which each entire sequence corresponds to a single gesture label. This setup is formulated as a sequence-level classification task, where the model receives the full frame sequence and predicts a single class. In contrast, Fig. 10(b) depicts an online (real-time) setting, where non-gesture segments are present and frame-wise labels may vary within a sequence. The model must therefore simultaneously perform gesture segmentation and classification in a streaming manner. This online scenario presents greater challenges, as the system is required to identify gesture boundaries (i.e., start and end frames) and assign the correct class labels to each detected segment. Additionally, because the number of gestures per sequence is unknown in advance, evaluation must consider both detection and localization performance—incorporating metrics such as false positive rate and the Jaccard index.

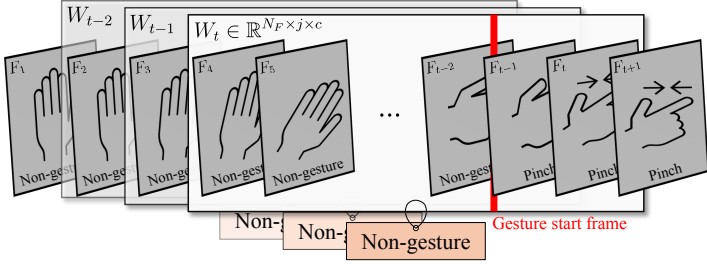

Figure 11: Window-based problem formulation for online gesture recognition. A continuous frame stream is partitioned into fixed-length windows ($W_t$), where aggregated frame labels determine the window label by majority voting. For windows containing both gesture and non-gesture frames, gesture boundaries are additionally labeled for localization.

As shown in Fig. 11 and discussed in Section 3.1, in an online setting, the incoming frame stream is partitioned into fixed-length windows $W_t \in \mathbb{R}^{N_F \times j \times c}$, where each window contains $N_F$ frames of skeleton data with $j$ joints and $c$ coordinate dimensions. A window-level label is determined by majority voting over the frame-level annotations within that window. In the case of "mixed" windows—i.e., windows containing more than one gesture-related label, such as both gesture and non-gesture frames—the gesture's start and end positions are additionally annotated to facilitate boundary localization. For such windows, the start index is initially set to 0 and the end index to $N_F - 1$, under the assumption that the gesture spans the entire window. Then, the label sequence is

traversed in temporal order to identify the actual transition points: when a frame label switches from non-gesture to a gesture class, the corresponding index is updated as the gesture start; conversely, when the label changes from a gesture class to a non-gesture label, the index just before the transition is assigned as the gesture end. If no transitions are found (i.e., the window contains only gesture or only non-gesture), the default start and end indices remain unchanged. Finally, the start and end indices are normalized to the range [0,1] by dividing by $N_F - 1$, resulting in two real-valued outputs, $g_s$ and $g_e$, for boundary regression.

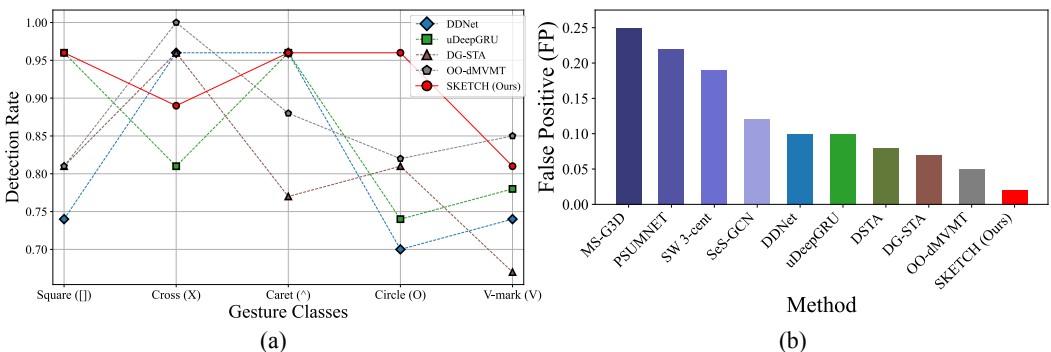

Figure 12: Performance comparison on the SHREC'19 dataset. (a) Per-class detection rate (DR) for the Square, Cross, Caret, Circle, and V-Mark gestures. (b) Total number of false positives (FP) aggregated across all gestures.

As shown in Fig. 12, the SHREC'19 dataset—where the training and test sets are drawn from entirely different groups of subjects, with only four subjects in the training set and nine unseen subjects in the test set—presents a substantial domain gap. This challenging setting demonstrates that our SKETCH method outperforms other benchmark approaches in both detection rate and false positive rate. These results indicate that our method robustly extracts high-level features that generalize well to unseen users and new domains.

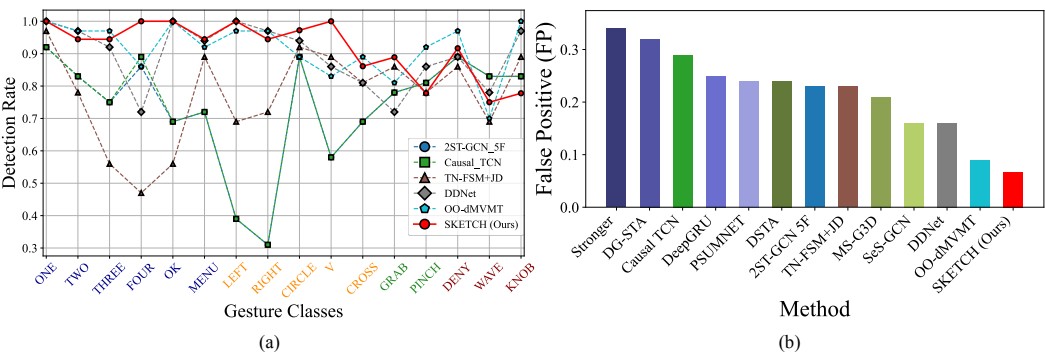

Figure 13: Performance comparison on the SHREC'22 dataset. (a) Detection rate (DR) for each of the 16 gesture classes. (b) Overall false positive (FP) aggregated across all classes.

Figure 13 illustrates that on the SHREC'22 benchmark—which features a larger set of gesture classes than SHREC'19 and, unlike SHREC'19, does not clarify whether the training and test subjects differ—our SKETCH method achieves a detection rate comparable to the state-of-the-art OO-dMVMT model, while significantly reducing the false positive rate. This reduction is particularly important for online gesture recognition. These results position SKETCH as the new state-of-the-art approach in the field.

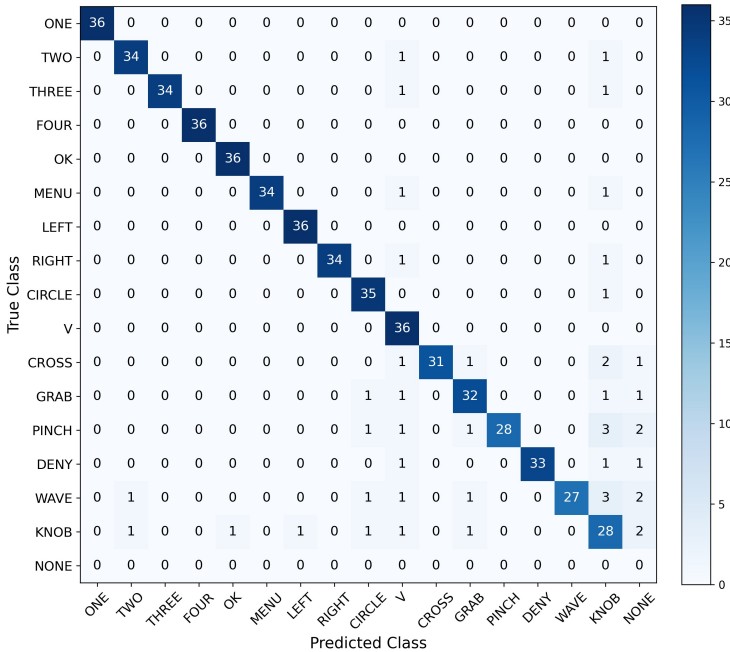

Figure 14: Confusion matrix ($17 \times 17$) on the SHREC'22 dataset. Each row indicates ground-truth and each column shows predicted class, including the additional NONE class to reflect non-gesture.

Figure 14 visualizes the confusion matrix for SHREC'22, where each of the 16 gesture classes is evenly represented by 36 instances, and shows that our method achieves uniformly high performance across all classes.

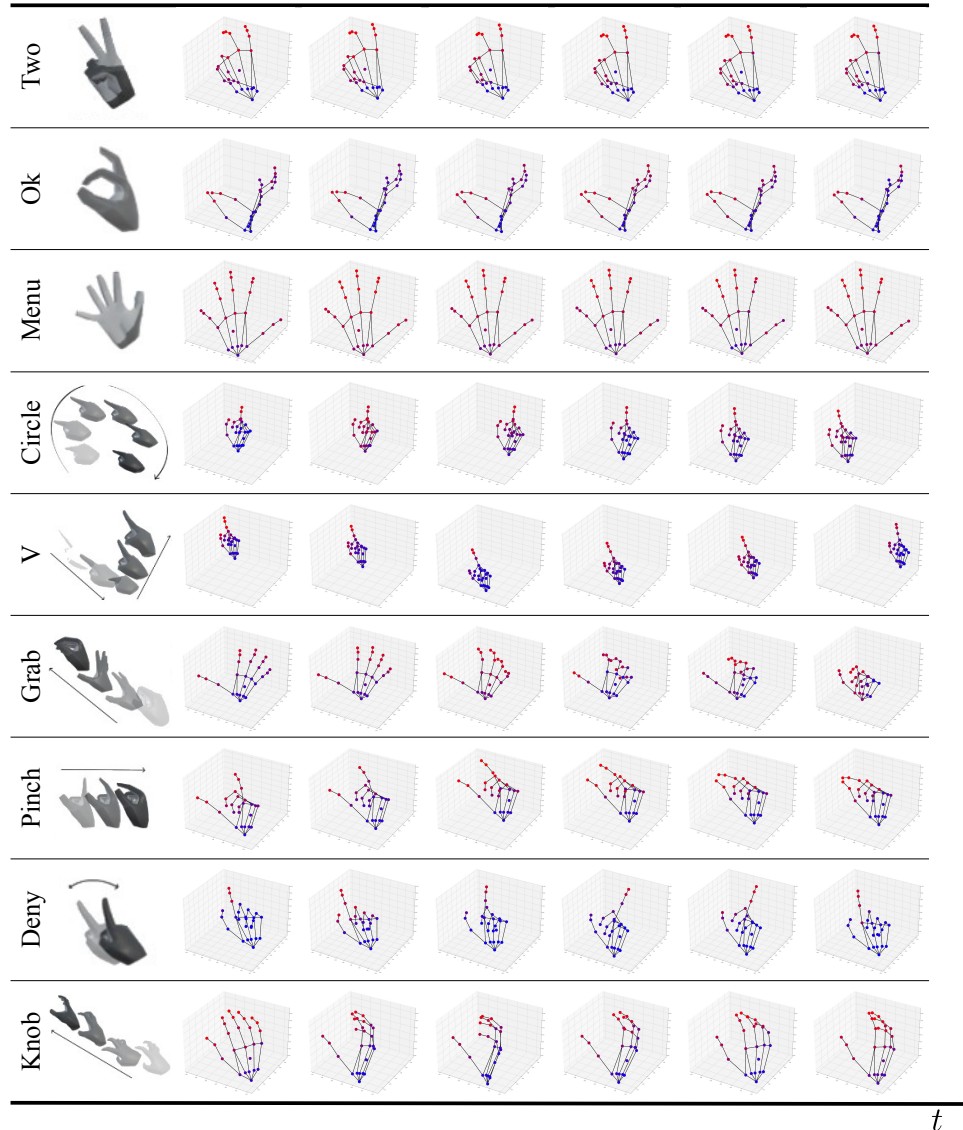

Figure 15: Extended visualization of joint-level attention (corresponding to Fig. 5) for gestures in the SHREC'22 dataset.

To investigate whether the model distinguishes gestures based on meaningful semantic cues, we applied Grad-CAM to the SKETCH-based 2D graph images and visualized attention scores for each joint. The Grad-CAM activations were first extracted from the 2D image space, and each joint's attention value was mapped back to its physical 3D coordinates ($x$, $y$, and $z$). Attention values across axes were then averaged to obtain a scalar score per joint. The attention maps (Fig. 15) show that for the Ok gesture, the model concentrated on the thumb and index finger as well as the fingertips of the remaining three fingers; for the Grab gesture, high attention was uniformly distributed across all five fingers; and for the Deny gesture, the focus was exclusively on the index finger. These findings demonstrate that the model relies on critical joints for gesture discrimination, closely mimicking human perception. This behavior further provides evidence that the model captures high-level semantic structure and avoids overfitting to low-level geometric features, validating the SKETCH-based representation in terms of both interpretability and effectiveness.

**Algorithm 1** Online Gesture Recognition Algorithm

---

**Require:** Model $f$, total number of gesture classes $K$, non-gesture label $C_{non}$, window size $N_F$, queue size $Q$, threshold value $\tau$ (frames; for minimum gesture length in SHREC'19).

**Input:** Stream of predictions, $W_t = \{X_1, X_2, \ldots, X_{N_F}\}$, where $t$ is the current time index, $X \in \mathbb{R}^{j \times 3}$, and where $j$ is the number of joints.

**Output:** Gesture start frame, end frame, and class label.

1:   $t \leftarrow 1, start \leftarrow -1, end \leftarrow -1$
2:   $prev\_g \leftarrow C_{non}$                                         ▷ Initialize previous majority vote result
3:   $hist \leftarrow \{0, 0, \ldots, 0\} \in \mathbb{R}^{1 \times K}$                          ▷ Histogram of gesture class occurrences
4:   $Queue \leftarrow \{C_{non}, C_{non}, \ldots, C_{non}\} \in \mathbb{R}^{1 \times Q}$   ▷ Sliding window (e.g., using deque for efficient updates)
5:   **while** receiving new predictions $W_t$ **do**
6:       **if** size of $Queue = Q$ **then**
7:           Remove oldest element from $Queue$
8:       **end if**
9:       $\hat{y}_t \leftarrow f(W_t)$                                       ▷ Predict label from model
10:      Append $\hat{y}_t$ to $Queue$
11:      $g_t \leftarrow \arg\max(\text{bincount}(Queue))$ ▷ Majority vote in the window In case of tie, choose the label with the smallest index
12:      **if** $g_t \neq C_{non}$ **then**                              ▷ Gesture detected
13:          **if** $prev\_g = C_{non}$ **then**                    ▷ Gesture start detected
14:             $start \leftarrow \max(1, t - \lfloor N_F/2 \rfloor)$
15:          **end if**
16:          $hist[g_t] \leftarrow hist[g_t] + 1$
17:      **else**                                          ▷ Non-gesture detected
18:          **if** $prev\_g \neq C_{non}$ **then**                   ▷ Gesture end detected
19:             $end \leftarrow \max(1, t - \lfloor N_F/2 \rfloor)$
20:             $chosen\_label \leftarrow \arg\max(hist)$ ▷ Final gesture label; tie-breaking similar as above.
21:             **if** SHREC'19 **then**                   ▷ Only one gesture in a sequence
22:                **if** $(end - start) \geq \tau$ **then**            ▷ Longer than threshold $\tau$
23:                  **Output** $(start, end, chosen\_label)$
24:                  **break**                  ▷ End processing for single-gesture sequence
25:                **else**
26:                  **skip**                     ▷ Too short; do not output
27:                **end if**
28:             **else**                           ▷ For diverse gestures in a sequence
29:                **Output** $(start, end, chosen\_label)$
30:             **end if**
31:             Reset: $start \leftarrow -1, end \leftarrow -1, hist \leftarrow \{0, 0, \ldots, 0\}$
32:          **end if**
33:      **end if**
34:      $prev\_g \leftarrow g_t$                                      ▷ Update previous vote
35:      $t \leftarrow t + 1$
36: **end while**

---

# G    Training Strategies and Implementation Details

All experiments were conducted using the AdamW optimizer [29] with a weight decay of $1 \times 10^{-4}$, a batch size of 32 trained over 20 epochs, and loss weights $\lambda_1 = \lambda_2 = \lambda_3 = 1$. We applied a linear warm-up [14] from 0 to the target learning rate over the first $30\%$ of training steps, followed by a cosine-annealing decay schedule.

For the SHREC'19 dataset, the initial learning rate was set to $8 \times 10^{-5}$, the window length $N_F$ to 40, the queue length $Q$ was set to nine, and the threshold $\tau$ (see in Algorithm 1) was set to 21, resulting in a delay of four frames in our Swin [27] Large model.

For SHREC'22, we retained these core settings but reduced the initial learning rate to $4 \times 10^{-5}$ and set $N_F$ to 16, and the $Q$ was set to 18, resulting in a delay of eight frames in our Swin Large model. To improve label reliability while training, we retained only those training windows in which the majority gesture occupied at least $70\%$ of the $N_F$ frames; windows lacking such a dominant class were discarded because their mixed labels reduced annotation confidence.

For the input resolution of $224 \times 224$ used in the pretrained backbones (Tabs. 1 and 2), we first generated $225 \times 225$ images through SKETCH and then applied center cropping to obtain the final $224 \times 224$ inputs.

For both SHREC'19 and SHREC'22, we applied the same data augmentation strategy during training. Specifically, augmentation was applied only to the training set, while the test set remained unchanged. Each temporal window was augmented using a random 3D rigid rotation, where the Euler angles $(r_x, r_y, r_z)$ were sampled from a uniform distribution $U(-\theta, \theta)$ with $\theta = 0.3$ radians. The sampled rotation was applied consistently to all frames within the window, preserving temporal dynamics and skeletal topology while introducing viewpoint variations. For each original window, two additional augmented windows were generated, resulting in a threefold increase in the training data. In Tabs. 1 and 2, the notation 'Aug.' was used exclusively for our method to indicate that the proposed augmentation was applied, whereas other methods were reported with their original training protocols. For our method, configurations without this notation ('Aug.') explicitly indicate training without augmentation.

# H  Toy Example Evaluation of Geometric Shortcut

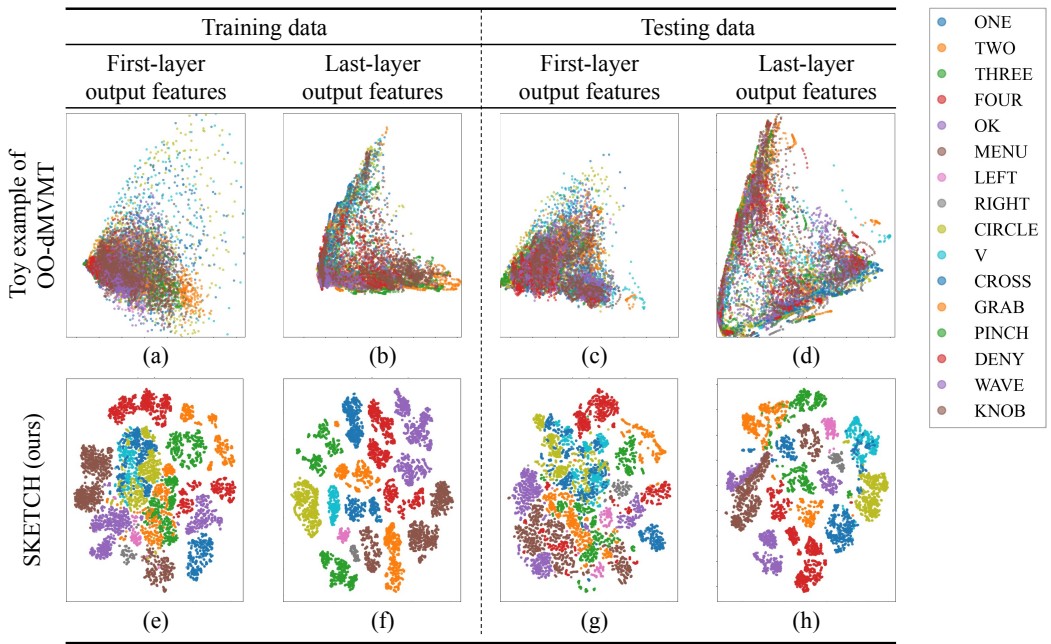

Figure 16: t-distributed stochastic neighbor embedding (t-SNE) visualizations of model representations on SHREC'22. (a–d) toy example of OO-dMVMT; (e–h) our SKETCH. (a, e) and (b, f) illustrate t-SNE visualizations of the first- and last-layer output features on the training data, respectively, while (c, g) and (d, h) present the corresponding features on the testing data. Each point is colored by its gesture class as shown in the legend.

This experiment was conducted to demonstrate the geometric shortcut phenomenon that arises when raw skeleton data is used as input. We implemented a toy example using the OO-dMVMT architecture—the state-of-the-art model—replacing its hand-crafted features with raw skeleton coordinates. We evaluated this toy example on the SHREC'22 benchmark dataset using only pure windows—those in which all ($N_F$) frames share the same label—excluding mixed label windows to avoid confusing feature clusters. Figure 16 depicts t-distributed stochastic neighbor embedding (t-SNE) plots of every feature representation for pure windows from both the training and testing data. For each of the raw-input OO-dMVMT toy example and our SKETCH model, we applied t-SNE to the features extracted by the first and the last layers of the model. Specifically, for SKETCH the first-layer is the zeroth Swin Transformer block, whereas for the OO-dMVMT toy example they correspond to the features at which multi-scale features are concatenated, (see [8], which introduces this process). In both models, the output of the final layer corresponds to the high-level feature representation just before it is passed to the classifier head. Under these conditions, the OO-dMVMT toy example achieved 96.22% window-level training accuracy and 89.82% window-level test accuracy before post-processing, whereas our proposed SKETCH model attained 99.00% and 92.58%, respectively.

Despite its reasonably high window-level accuracy, the toy example of OO-dMVMT still falls into the geometric shortcut phenomenon. This can be observed through the comparison in Fig. 16(d) and (h), which present t-SNE visualizations of the last-layer output feature representations for OO-dMVMT and SKETCH, respectively. Compared to SKETCH, the toy example of OO-dMVMT produces features that are merely hyperplane-separable—i.e., sufficient for a linear classifier to distinguish—but fail to form clearly clustered distributions by class, highlighting the lack of semantically meaningful structure. This reflects the toy example of OO-dMVMT's tendency to exploit geometric shortcut—i.e., relying on low-level, coordinate-based cues that lead to non-generalizable decision boundaries—rather than learning semantically rich, well-separated class clusters. This indicates that feeding raw skeleton coordinates causes the OO-dMVMT model to overfit to trivial coordinate cues—extracting only

low-level features instead of semantically meaningful, high-level representations—and thus provides evidence of the geometric shortcut effect.

# I  Impact of Spatial Emphasis in Skeleton-Based Gesture Recognition

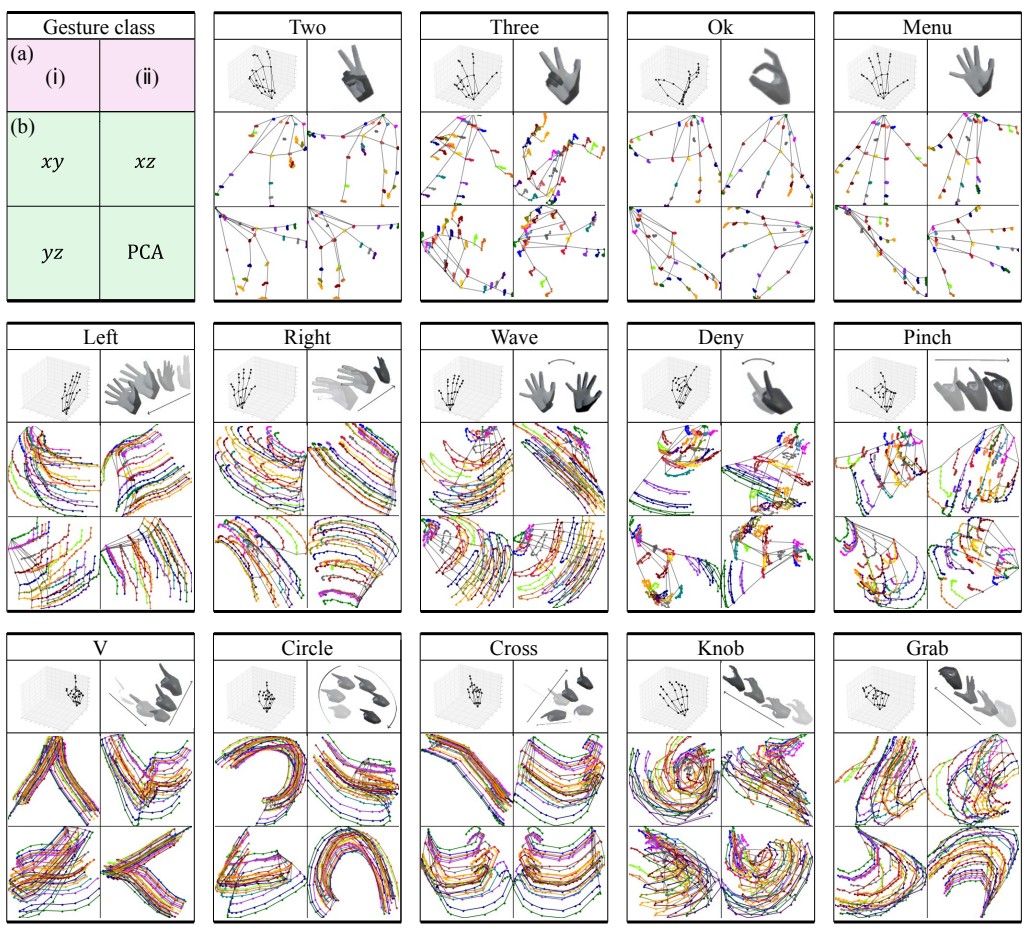

Figure 17: Visualization of 14 representative gesture classes from the SHREC'22 dataset using the ShapeProj plotting method. Each block corresponds to a gesture class. Pink section (a) shows (i) the 3D skeleton and (ii) the reference image. Green section (b) presents four projection views of the joint trajectories: projections onto the $xy$, $xz$, and $yz$ planes, and a view generated via principal component analysis (PCA). This layout highlights the spatial configurations captured by ShapeProj across multiple perspectives.

Table 13: Classification performance comparison between ShapeProj and SKETCH on the SHREC'22 dataset. Backbone configurations (e.g., S-B-4-12-384) follow the same notation as described in Tab. 1.

| Method | DR↑ | FP↓ | JI↑ |
|---|---|---|---|
| ShapeProj (S-B-4-12-384) | 0.82 | 0.05 | 0.79 |
| ShapeProj (S-L-16-384) | 0.85 | 0.05 | 0.81 |
| SKETCH (S-B-4-12-384) | 0.91 | 0.03 | 0.86 |
| SKETCH (S-L-16-384) | 0.92 | 0.02 | 0.87 |

Certain hand gestures in 3D space exhibit highly distinctive and discriminative shapes, such that the gesture class can be identified solely based on the spatial configuration of joints. As shown in Fig. 17(a), gestures such as Two, Ok, and Pinch display clear morphological distinctions: Two involves extension of only the index and middle fingers; Ok is formed by creating a circle with the thumb and index finger while the remaining fingers are extended; and Pinch is characterized by the thumb and index finger closing together with the other fingers naturally curled inward. These clearly defined hand shapes serve as intuitive cues for human perception and can also act as effective features for gesture recognition models.

Building on this observation, this section investigates how an alternative plotting strategy—referred to hereafter as ShapeProj—affects model performance, with a particular focus on preserving the natural spatial structure of the human hand in 3D space through orthogonal projection. While both ShapeProj and SKETCH approaches utilize raw coordinate data without relying on rule-based feature engineering (i.e., hand-crafted features), the proposed method adopts a rendering technique designed to maintain the geometric integrity of the hand. We aim to assess whether visualizations that preserve the physical configuration of the hand (as shown in Fig. 17(b)) can serve as effective representations for gesture recognition.

ShapeProj projects the 3D skeleton sequence ($\in \mathbb{R}^{N_F \times j \times 3}$) onto four views: the $xz$, $xy$, and $yz$ planes, and an additional axis on which the variance is maximized when the 3D skeleton sequence is projected (derived by principal component analysis (PCA)). The use of PCA aims to depict the hand shape from a perspective that most clearly separates joint positions, regardless of the absolute location of the gesture in 3D space, and with minimal loss of information. For visualization, the resulting 2D joint coordinates obtained from each orthogonal projection and PCA are assigned to the four quadrants of a single image. Prior to plotting, min–max normalization is applied independently to each 2D axis of the data ($\in \mathbb{R}^{N_F \times j \times 2}$) to ensure scale consistency. Each joint is assigned a unique color, and the temporal trajectory of each joint is illustrated by connecting its positions across frames using colored lines. Markers are added to indicate observed joint positions, and for the first frame in each quadrant, black edges are drawn to connect joints based on semantic hand structure, enabling direct rendering of the hand shape.

Figure 17(b) shows the visualization results for the SHREC'22 dataset using ShapeProj. Compared to SKETCH, this approach more clearly represents the spatial configuration of the hand joints in 3D space, but sacrifices temporal resolution as a trade-off. As shown in Tab. 13, the classification performance of the hand shape-preserving visualization (ShapeProj) is sufficiently robust to suggest that emphasizing spatial configuration can provide meaningful discriminative cues for gesture recognition.

However, comparison with SKETCH reveals that SKETCH-based representations consistently outperform those generated by ShapeProj. This suggests that for an online, streaming-based gesture recognition task, the model relies more heavily on relative motion patterns between joints over time, rather than on spatial configurations in which temporal dynamics are less prominently expressed. Such a tendency is also observed in Tab. 7, where we compare hand-crafted features by plotting joint collection distances (JCD) and frame difference vectors (FD). JCD captures static spatial structure by computing distances between joints within the same frame, while FD encodes temporal dynamics by measuring displacement across frames. Despite both being hand-crafted, the FD-based images yield superior performance compared to JCD, highlighting the importance of temporal information. In summary, while the spatial configuration of the hand provides a strong and informative basis for classification, our experiments indicate that characteristic motion patterns between joints over time play a more decisive role in effectively solving the gesture recognition task.

