# OpenReview forum: "Doodle to Detect: A Goofy but Powerful Approach to Skeleton-based Hand Gesture Recognition"
_NeurIPS.cc/2025/Conference — NeurIPS 2025 poster_

### Official Review · Reviewer_sCbL · 2025-06-23

**Clarity:** 3
**Significance:** 2
**Originality:** 2
**Rating:** 4
**Confidence:** 4

**Summary:**

This paper presents a hand gesture recognition algorithm using 3d skeleton sequences as input. It works by transforming the collection of sequences (x,y,z)_t for all the joints of the hand into three graph plots {x(t),y(t),z(t)}, where for each coordinate, all the joints are represented within the same plot, with different colours, and using min-max normalisation. To retrieve the different relative dynamic ranges between coordinates, the same data are fed to a module that outputs a range embedding added to each plot. The three resulting plots - for a fixed-sized temporal window - are vertically stacked to form an image that is submitted to a swin-transformer network with one classification head that performs the gesture recognition, and two regression heads that respectively locate the start and end times of the gesture.

**Questions:**

The role of the Dynamic Range Embedding is not clear. Additional explanations are needed. And why learn an embedding range and not simply compute it from the sequences?
One important point of gesture recognition methods working on pose data is their robustness to pose estimation errors. How does your method deal with that?

**Ethical Concerns:**

["NO or VERY MINOR ethics concerns only"]

**Final Justification:**

I have read the reviews of my colleagues and the author's rebuttal.

After consideration of the comprehensive point-to-point anwsers that have been provided by the authors, and although I'm still reserved about the method, and I find that some arguments - like those about the limitations of the analytical normalisation - are a little weak, I recognise the considerable efforts that have been made by the authors to improve their paper, and my opinion is significantly more positive.

I therefore increase my score to "borderline accept".

Best regards,

**Limitations:**

There is a short section on limitations, but I consider that it does not address the main limitations mentioned above: 1) what is the fundamental contribution of the proposed visual representation of skeletal time series? 2) what is the robustness of the method with respect to pose estimation errors?

**Paper Formatting Concerns:**

The formatting seems globally OK, maybe there is too many things in the supplementary material (and too many reference to it in the main paper).

**Quality:**

2

**Strengths And Weaknesses:**

The paper is well written and easy to read. The authors made a remarkable effort in the graphical presentation of their paper and the figures are excellent.
The major problem I have in this paper lies precisely in its visual representation of the skeletal data. I find it not only "goofy" as confessed in the title (and only there curiously), I find it scientifically shocking. The argument of "reduction" from 4d to 2d data does not make sense to me, since it's about transforming a light collection of time series into a high definition image with sparse relevant information (that lose topological information between joints BTW). And I don't think that good results are sufficient to make it acceptable. The authors should at least compare their approach with a more natural method using the native time series, and be able to explain why this anthropomorphic stacked-plots-as-image classification method possibly works better.
I also do not agree with some concluding claims saying that "These results suggested that skeleton-based hand gesture recognition can play a key role", and that "the SKETCH module opened up a new perspective in hand gesture recognition through an interpretable data representation." I consider that interpretability comes from the pose estimation and the skeletal input, not from your method.

---

> ### Author Rebuttal · Authors · 2025-07-31
>
> To ensure a complete and thoughtful response within the space constraints, **we kindly refer the Reviewer to tables cited in responses to other Reviewers**, as several analyses are shared across comments. **We sincerely appreciate your understanding.**
>
> **Disagreement with the 4D-to-2D Reduction Perspective**
> - We appreciate the Reviewer’s comment and agree that our method is not a true dimensionality reduction. Rather, it aims to unlock and structurally reorganize the rich spatio-temporal patterns embedded in the original 4D skeleton data—patterns that are often underutilized by conventional hand-crafted feature approaches.
> - As detailed in **Appendix A**, directly using raw joint coordinates risks overfitting due to geometric shortcuts and coordinate bias, which normalization alone cannot fully resolve. SKETCH addresses this by transforming skeleton sequences into 2D graph images that highlight joint trajectories and temporal structure, enabling the model to learn more robust motion features.
> - While this incurs some additional computational cost (**Tab. A** in our response to **Reviewer JPRk**), we believe the improved accuracy and real-time performance justify the trade-off.
>
> **Fundamental contribution of the proposed visual representation**
> - Prior work and limitations
>    - Please refer to our response to **Reviewer JPRk** under “Limitations of Prior Skeleton Methods and SKETCH’s Solutions” for details on related work and challenges.
>  - Why visualization is needed and how SKETCH addresses existing limitations
>    -  Addresses **① raw data** challenges:
>       1. By transforming raw coordinates into images, SKETCH uses pretrained visual backbones to extract high-level features, preventing overfitting to geometric cues and reducing reliance on shortcuts.
>       2. Coordinate bias is mitigated through the pretrained backbone’s invariance to spatial transformations, enabling gesture recognition despite changes in viewpoint, scale, or location.
>       3. Inter-subject variability is addressed through per-window normalization, using relative joint positions in image space.
>    -  Overcomes Limitations of **② Hand-Crafted Features**:
>       4. Our representation eliminates the need for expert design, adopting a transformation similar to how humans interpret visual charts, making it generalizable and intuitive.
>       5. SKETCH enables interpretability by converting raw skeletons into visual images, allowing inspection of joint and temporal segment contributions to classification (**Figs. 4, 7 (Appendix D), and 14 (Appendix F)**).
>       6. Once mapped to the image domain, our data can be readily processed by global self-attention backbones, enabling explicit modeling of long-range dependencies between joints.
>  - Contributions of the work
>    - The proposed SKETCH method leverages raw skeleton data to enhance interpretability and preserve rich spatio-temporal information, resulting in improved recognition performance compared to prior approaches (**Tabs. 1 and 2**). Under these conditions, SKETCH shows fast convergence in small number of epochs ($≤ 20$ epochs) (**Tab. B** in our response to **Reviewer JPRk**) and high generalization (**Tabs. C and D** in our response to **Reviewer JPRk**).
>
> **Concern About Loss of Joint Topology**
> - While we agree that topology is an important cue, SKETCH deliberately omits explicit 3D topological structure, it was deliberately designed to emphasize temporal joint dynamics via structured, time-aligned graph images.
> - As shown in **Appendix E.3** and **Supp. C**, we compared spatial (e.g., JCD) and temporal (e.g., FD) features under the same SKETCH pipeline; temporal ones consistently outperformed spatial (**Tab. 7 (Appendix E.3)**), suggesting motion cues are more discriminative in our setting.
> - For further analysis, we developed ShapeProj—a topology-preserving visualization designed purely for analysis—and found SKETCH yielded better accuracy (**Tab. 11 (Supp. C)**). This suggests temporal kinematics play a more decisive role in solving gesture recognition tasks.
> - Additionally, SKETCH’s topology-agnostic design simplifies deployment across diverse skeleton datasets requiring no prior structural knowledge, as further discussed in our response to **Reviewer D9gv's Question 2**.
>
> **Comparison with Native Time-Series Methods:**
> - Thank you for pointing this out. We clarify the model types used in **Tabs. A and B**, and now provide explicit comparisons to native time-series baselines (RNN, TCN, and Transformer).
>
> **Table A: Results on SHREC'19 compared to native time series methods**
> |Methods|Architecture|DR↑|FP↓|
> |-|:-:|-|-|
> |DSTA [40]|Transformer|0.81|0.08|
> |DG-STA [5]|Transformer|0.81|0.07|
> |uDeepGRU [4]|GRU (RNN)|0.85|0.10|
> |Ours|SKETCH|0.92|0.02|
>
> **Table B: Results on SHREC'22 compared to native time series methods**
> |Methods|Architecture|DR↑|FP↓|JI↑|
> |-|:-:|-|-|-|
> |DeepGRU [29]|GRU (RNN)|0.26|0.25|0.21|
> |DG-STA [5]|Transformer|0.51|0.32|0.40|
> |DSTA [40]|Transformer|0.73|0.24|0.61|
> |TN-FSM+JD [12]|Transformer|0.77|0.23|0.63|
> |CausalTCN [12]|TCN|0.80|0.29|0.68|
> |Ours|SKETCH|0.92|0.07|0.87|
>
> - This highlights SKETCH’s strength in capturing temporal dynamics over native time-series methods.
>
> **Advantages of the Stacked Plots as Image Representation:**
> - As discussed in **Appendix A**, raw joint sequences often suffer from coordinate bias and geometric shortcuts, even with normalization. SKETCH mitigates these issues by converting sequences into structured graph images that emphasize consistent joint trajectories and temporal dynamics.
> -  This allows pretrained vision transformers to better capture motion patterns and focus on relevant attention cues, yielding more robust and interpretable learning than approaches relying on hand-crafted features.
>
> **Clarifying the Source of Interpretability**
> - We strongly agree that interpretability stems from raw skeletal inputs and apologize for the unclear phrasing.
> - SKETCH enables intuitive joint-level attention by preserving alignment with original joint movements. In contrast, when applied to hand-crafted features, interpretability diminishes, as these features do not preserve joint-specific semantics and often result in unintuitive attention patterns.
> - SKETCH mitigates geometric shortcuts caused by using raw coordinates directly by transforming sequences into temporally structured graph images.
> - Our intention in referring to SKETCH as “interpretable” was to highlight that it maintains the transparency of raw skeleton inputs while enabling effective training in otherwise failure-prone settings.
>
> **Clarifying the Role of Dynamic Range Embedding**
> - We clarify that the Dynamic Range Embedding (DRE) is designed to address a limitation of the axis-wise min–max normalization used in SKETCH. While normalization improves consistency across subjects and viewpoints, it removes the relative magnitude of joint motion across axes—information that can be critical for distinguishing between gestures.
> - To restore this, DRE injects a scalar encoding of the original per-axis dynamic range into each subplot of the graph image. Unlike fixed statistical encodings (e.g., variance or max–min), DRE is learnable, allowing the model to adaptively capture semantically meaningful motion scales during training. This idea draws inspiration from prior work showing that learnable positional embeddings—used in place of sinusoidal encodings in Transformers—improve pattern modeling and flexibility during fine-tuning (**Appendix E.5**) [g].
> - We also conducted ablation studies (**Tabs. C and D**) comparing DRE to fixed encodings. DRE achieved better performance, likely due to its robustness to outliers and its ability to optimize jointly with the model. This highlights DRE’s value in enhancing discriminative capacity while maintaining model robustness.
>
> **Table C: Effectiveness of DRE vs. statistical encodings (Std. / Max–Min) on SHREC'19**
> |Ablation|Std.|Max‑Min|DRE|DR↑|FP↓|
> |-|:-:|:-:|:-:|-:|-:|
> |SKETCH (w/o DRE)|✓|||0.89|0.02|
> |SKETCH (w/o DRE)||✓||0.89|0.03|
> |SKETCH (w/ DRE)|||✓|0.92|0.02|
>
> **Table D: Effectiveness of DRE vs. statistical encodings (Std. / Max–Min) on SHREC'22**
> |Ablation|Std.|Max‑Min|DRE|DR↑|FP↓|JI↑|
> |-|:-:|:-:|:-:|-:|-:|-:|
> |SKETCH (w/o DRE)|✓|||0.90|0.08|0.84|
> |SKETCH (w/o DRE)||✓||0.89|0.06|0.84|
> |SKETCH (w/ DRE)|||✓|0.92|0.07|0.87|
>
> **Robustness to Pose Estimation Errors**
> - To assess SKETCH’s robustness to pose estimation errors, we additionally evaluated it under missing-data perturbations on SHREC’19—introduced only at test time—to examine generalization under degraded pose inputs.
>     - **Missing-joint robustness**: Simulated joint-sensor failure by randomly deleting joints and measuring performance.
>     - **Occlusion robustness**: Simulated partial marker occlusions by masking joint coordinates over time and joints, then measuring performance.
>     - **Test noise robustness**: We injected white Gaussian noise on training joint coordinates and evaluated the model on test sequences corrupted with the same noise.
> - As shown in **Tabs. E and F**, we evaluated detection rate (DR) and false positives (FP) under these three perturbations. Despite the synthetic perturbations, SKETCH remained robust by capturing the spatio-temporal flow of joint kinematics, rather than relying on individual joint trajectories.
>
> **Table E: Robustness of our method under various perturbations of missing-joints**
> |Perturbations|Missing Rate|DR↑|FP↓|
> |-|-|-|-|
> |-|-|0.92|0.02|
> |Sequence|30%|0.91|0.02|
> |Sequence|50%|0.89|0.02|
> |Sequence|70%|0.85|0.02|
> |Sequence|90%|0.73|0.34|
> |Frame|30%|0.90|0.02|
> |Frame|50%|0.78|0.20|
> |Frame|70%|0.06|0.00|
>
> **Table F:  Robustness of our method under noise perturbations with varying noise rates (std.)**
> |Perturbations|Std.|DR↑|FP↓|
> |-|-|-|-|
> |Noise|0.02|0.89|0.04|
> |Noise|0.03|0.89|0.04|
> |Noise|0.05|0.76|0.07|
>
> [g] Yu et al., "Lape: Layer-adaptive position embedding for vision transformers with independent layer normalization.", ICCV 2023.

---

> > ### Comment · Reviewer_sCbL · 2025-08-04
> >
> > Dear Authors,
> >
> > thank you for your detailed answers.
> >
> > I have read the other reviewers' comments and appreciate your effort in the point-by-point response you provide to all.
> >
> > About your rebuttal to my own concerns, I acknowledge that it answers some of them, but I am still not convinced in some particular points:
> >
> > - **Criticism about the conversion from time-series to stacked plot images:** I remain extremely unenthusiastic scientifically with the idea of converting a light collection of coordinate time series to a high resolution image; it might be a reduction of dimension, but definitely not of memory. Now I accept the fact that this is a purely personal feeling, clearly not shared by my review colleagues.
> >
> > - **Comparison with native time-series methods:** Thank you for answering this point and providing a comparison between your method and time-series approaches on SHREC'19 and SHREC'22. However I do not agree with the rationale you provide for your approach based on the limitations of time series, that claims that coordinates time-series methods necesseraly suffer from biases towards absolute coordinates that make them both view- and subject- dependent. It is evident that the naive normalisation methods presented in your Appendix A do not solve the problem. However there exist simple analytic solutions to address view-dependence, e.g. by setting the reference frame origin at the centre of the wrist in the initial frame, and subject-dependence, e.g. by normalising the edges with respect to a reference hand. Such analytical normalisation would be both simpler and *more explicable* than your method. Ideally, your comparison with time series methods, as well as your ablation on the Dynamic Range Embedding should take this into consideration.
> > - **Importance of Dynamic Range Embedding:** Thank you for this explanation and additional ablation study. I still believe though, as mentioned above, that DRE should be compared to pose- and subject- invariant analytical normalisation.
> > - **Considerations about Topology-agnosticity and Robustness to pose estimation errors:** Thank you for providing these elements. I believe that the hypotheses that the "drawing-plots" method could be more robust than direct methods, precisely because they ignore the inherent graph structure and then are (1) able to generalise well to any skeleton structure and (2) more resilient to missing node or pose estimation errors, is *way more* convincing than the arguments of coordinate bias and geometric shortcuts. Since this is supported by your experiments, I suggest that you shoud put more emphasis in these points.
> >
> > Despite my reservations, I will take into account the positive elements provided by the authors in this rebuttal to update my score.
> >
> > Best regards,

---

> ### Author Response · Authors · 2025-08-05
>
> Dear Reviewer,
>
> Thank you for responding to our rebuttal.
> We would like to respond to the four points.
>
>
> - Criticism about the conversion from time-series to stacked plot images:
>    - We agree with the Reviewer’s pointed comment that ‘it might be reduction of dimension, but definitely not of memory.’ The phrase “transforms four-dimensional (time, $x$, $y$, and $z$) skeleton raw data into a simple and intuitive visual representation” in the main text does not imply scientific dimensionality reduction (dimensionality reduction) but rather indicates a change in **format** from four-dimensional coordinate time-series to two-dimensional graph image. Through the transformation process via SKETCH, we mitigate the **geometric shortcut and coordinate bias (subject-, view-variant)** that occur in raw data under **naïve normalisation** by leveraging a pretrained backbone–based high-level semantic feature extractor.
>
>
> - Comparison with native time-series methods:
>    - We agree with the **Reviewer** that analytical normalisation alleviates view- and subject-dependency. However, analytical normalisation typically relies on a **reference joint**, which may introduce the following limitations:
>       1. It implies the need for **domain expert knowledge** (or prior topology information refer to **Section 1.**) for a specific sensor or dataset (e.g., in the task of human action recognition under the Microsoft Kinect for Windows V2 Skeleton **[h]**, Spine-based is at index 0).
>       2. The reference joint may (a) **have different indices depending on the sensor or framework** (e.g., Spine-Base is at index 0 in Microsoft Kinect V2 Skeleton, whereas the MidHip is at index 8 in OpenPose **[i]** BODY 25 key points) or (b) **not exist** (e.g., COCO 2D **[j]** does not have a central Spine-Base or MidHip index).
>       3. In real-world scenarios, **missing or noisy reference joint** causes degraded generalisation performance, leading to **reference joint-dependency**.
>    - In contrast, SKETCH is **robust** to these issues as mentioned in **Reviewer’s [Comment 4]**. We will place clearer emphasis on this robustness in the revised manuscript, as suggested by the Reviewer.
>
>
> - Importance of Dynamic Range Embedding:
>    - Reflecting the **Reviewer’s suggestion**, we will conduct an **ablation study** applying analytical normalisation to more precisely analyze the performance of DRE:
>       1. Vanilla Transformer + naïve normalisation
>       2. Vanilla Transformer + analytical normalisation
>       3. Vanilla Transformer + analytical normalisation + DRE
>    - We will organize the results and provide a response promptly upon completion.
>
>
> - Considerations about Topology-agnosticity and Robustness to pose estimation errors:
>    - We agree with the **Reviewer’s comment** that “precisely because they ignore the inherent graph structure and then are (1) able to generalise well to any skeleton structure and (2) more resilient to missing node or pose estimation errors.”
>    - While our method also addresses the limitations of naïve normalisation, we will revise the manuscript to more clearly clarify that our approach mitigates the geometric shortcuts and coordinate bias introduced by naïve time-series normalisation. In addition, as suggested by the Reviewer, we will place greater emphasis in the main text on (1) generalisation performance independent of sensors or specific datasets, and (2) robustness to missing nodes or pose estimation errors, which are key benefits arising from the inherent graph structure (topology)-agnosticity of our approach.
>
>
> I am confident that incorporating the Reviewer's comments will greatly enhance the overall quality of the paper. Thank you sincerely for your invaluable feedback.
>
>
> [h] H. Alabbasi et al., "Human motion tracking & evaluation using Kinect V2 sensor," 2015 E-Health and Bioengineering Conference (EHB), Iasi, Romania, 2015, pp. 1-4, doi: 10.1109/EHB.2015.7391465.
>
> [i] Cao, Zhe, et al. "Openpose: Realtime multi-person 2d pose estimation using part affinity fields." IEEE transactions on pattern analysis and machine intelligence 43.1 (2019): 172-186.
>
> [j] Cao, Zhe, et al. "Realtime multi-person 2d pose estimation using part affinity fields." Proceedings of the IEEE conference on computer vision and pattern recognition. 2017.

---

> ### Author Response · Authors · 2025-08-06
>
> Dear Reviewer,
>
> Thank you again for your thoughtful and constructive feedback. As previously mentioned, we conducted additional experiments to further explore the efficacy of **analytical normalisation (AN)** and our proposed **Dynamic Range Embedding (DRE)**.
>
> ---
> **1. Limitations of Naïve Normalisation**
>
> As shown in **Tab. G**, training a vanilla Transformer model on raw skeleton sequences using **naïve normalisation (NN)** results in notably limited performance. This setting suffers from **geometric shortcuts and coordinate bias**, leading the model to overfit to low-level geometric patterns (e.g., absolute hand heights) that do not generalise well across subjects or viewpoints.
>
> **Table G: Performance of vanilla Transformer with NN on SHREC’19 and SHREC’22.**
> |Dataset|DR↑|FP↓|JI↑|
> |:-:|:-:|:-:|:-:|
> |SHREC'19|0.41|0.14|-|
> |SHREC'22|0.68|0.16|0.61|
>
> ---
> **2. Effectiveness and Limitations of Analytical Normalisation**
>
> To empirically validate the benefits of view- and subject-invariant **AN**, we compared our method with existing approaches that explicitly adopt such strategies, including **BlockGCN** [58], **DS-GCN** [l], and **ProtoGCN** [b], on the SHREC’19 dataset (**Tab. H (i-iii)**).
>
>   - **BlockGCN**, **DS-GCN**, and **ProtoGCN**: Implements viewpoint invariance by converting all joints to relative coordinates based on a reference joint.
>
> These methods demonstrate the **effectiveness of AN**, and our experiments confirm that such strategies alleviate view- and subject-dependency as the Reviewer noted.
>
> However, we also emphasise that **AN may introduce its own limitations**:
>
>   - It often **requires domain expert knowledge**, such as selecting an appropriate reference joint based on prior topology information for a given dataset or sensor configuration.
>   - The definition and index of the **reference joint may differ across datasets** (e.g., SpineBase at index 0 in Kinect V2, MidHip at index 8 in OpenPose BODY-25) or **even be absent** (e.g., COCO 2D lacks a central reference joint).
>   - As the Reviewer highlighted, **real-world scenarios are prone to pose estimation errors**; if the reference joint is noisy or missing, **AN** can severely degrade performance due to its reliance on that joint.
>
> In contrast, our proposed SKETCH method is **topology-agnostic**—it does not rely on any specific reference joint or domain-specific preconditions —which enables **generalisation across diverse skeleton structures and provides resilience to pose estimation errors**, as it avoids dependencies that may be fragile in real-world scenarios with missing or noisy joints.
>
> **Table H: Performance comparison on SHREC’19 between SKETCH and methods employing AN.**
> |Index|Dataset|Method|Venue|AN|DR↑|FP↓|
> |-|:-:|:-:|:-:|:-:|:-:|:-:|
> |(i)|SHREC'19|BlockGCN|CVPR'24|✓|0.83|0.04|
> |(ii)|SHREC'19|DS-GCN|AAAI'24|✓|0.80|0.05|
> |(iii)|SHREC'19|ProtoGCN|CVPR'25|✓|0.86|0.05|
> |(iv)|SHREC'19|SKETCH|||0.92|0.02|
>
> ---
> **3. Analytical Normalisation with Vanilla Transformer Backbone**
>
> To further isolate the effects of normalisation, we applied **AN** schemes to a **vanilla Transformer backbone**, which is architecturally similar to SKETCH (without the image-based transformation). Results on SHREC’19 and SHREC’22 (**Tab. I (i) and (iii)**) confirm that:
>   - **AN** aimed at addressing view-dependence shows clear performance improvements even under this experimental setting.
>
> **Table I: Ablation study on SHREC’19 and SHREC’22 under AN to evaluate the performance impact of DRE.**
> |Index|Dataset|AN|DRE|DR↑|FP↓|JI↑|
> |-|:-:|:-:|:-:|:-:|:-:|:-:|
> |(i)|SHREC'19|✓||0.62|0.09|-|
> |(ii)|SHREC'19|✓|✓|0.64|0.09|-|
> |(iii)|SHREC'22|✓||0.81|0.15|0.72|
> |(iv)|SHREC'22|✓|✓|0.82|0.13|0.75|
>
> ---
> **4. Compatibility of DRE with Analytical Normalisation**
>
> We introduced **Dynamic Range Embedding (DRE)** to recover motion magnitude information lost during SKETCH’s axis-wise min–max normalisation. By learning and injecting the dynamic range of each axis into the visual representation, DRE helps retain scale-related cues important for distinguishing gestures. While a direct comparison between **DRE** and **AN** is not straightforward due to their differing objectives, we conducted a **proxy experiment** by applying **DRE** on top of **AN**—following the protocol used in **BlockGCN** [58]—within a vanilla Transformer backbone, and evaluated its impact on gesture recognition.
>
>   - See **Tab. I (i–ii)** for SHREC’19 results without and with **DRE**, and **Tab. I (iii-iv)** for corresponding results on SHREC’22.
>
> These results indicate that **DRE** provides complementary benefits even when **AN** is applied, by restoring axis-specific magnitude cues lost in the normalisation process.
>
> [b] Liu, et al. "Revealing key details to see differences: A novel prototypical perspective for skeleton-based action recognition." CVPR 2025.
>
> [l] Xie, et al. "Dynamic semantic-based spatial graph convolution network for skeleton-based human action recognition." AAAI 2024.

---

> ### Author Response · Authors · 2025-08-06
>
> We hope these additional analyses address your concerns. If any aspect remains unclear or if further clarification is needed, we would be grateful for your additional guidance.
>
> Best regards,

---

> > ### Comment · Reviewer_sCbL · 2025-08-08
> >
> > Dear authors,
> >
> > thank you for running those new experiments. The results are interesting, and I believe that incorporating them to the paper (and updating your Appendix A) will make your paper stronger. I also think that the justification for your "drawing plots" method will be more convincing when you add those new arguments.
> >
> > I then acknowledge your considerable efforts to improve your paper and I will increase my score at the end of the discussion phase.
> >
> > Best regards,

---

> > > ### Author Response · Authors · 2025-08-08
> > >
> > > Dear Reviewer,
> > >
> > > Thank you for your valuable comments.
> > >
> > > We sincerely appreciate the time and care you took in reviewing our manuscript, as well as your insightful and constructive suggestions that help strengthen the validity of our proposed method.
> > >
> > > In the revised manuscript, we will reinforce the persuasiveness of our approach by thoroughly addressing the points you raised, and we will incorporate the suggested relevant experimental results into Appendix A to further enhance the paper.
> > >
> > > If you have any further suggestions or feedback to help clarify the manuscript, we would be grateful to receive your input. We remain committed to refining the manuscript prior to final submission.
> > >
> > > Once again, thank you very much for your valuable time and thorough review.
> > >
> > > Best regards,

---

### Official Review · Reviewer_D9gv · 2025-07-03

**Clarity:** 3
**Significance:** 3
**Originality:** 3
**Rating:** 5
**Confidence:** 4

**Summary:**

This paper introduces SKETCH (Skeleton Kinematics Extraction Through Coordinated grapH), a novel approach for real-time hand gesture recognition using raw 3D skeleton data. Unlike conventional methods that rely on hand-crafted features, SKETCH transforms raw four-dimensional skeleton data (time, x, y, z) into interpretable graph images. These visual representations are processed by pretrained vision backbones (ViT/Swin Transformer) for gesture classification and boundary localization.

A central innovation is the Dynamic Range Embedding (DRE), which compensates for the loss of axis-relative scale caused by per-axis min–max normalization. The method also handles domain shifts—due to sensor viewpoints or user-specific kinematics—by applying normalization independently per window at inference time. SKETCH is evaluated on SHREC’19 and SHREC’22 benchmarks, achieving state-of-the-art accuracy and interpretability, while operating in real-time.

**Questions:**

1. Have the authors tried other backbones besides ViT and Swin Transformer, perhaps more lightweight ones? Because this paper addresses online recognition, efficiency is a critical factor that should be taken into account. And based on Table 1, this method doesn’t outperform existing methods in terms of time and FPS. So it would be helpful to provide some further analysis on efficiency.

2. Could the proposed SKETCH be applied to human body skeletons? The method doesn’t seem to restrict itself to any specific type of skeleton topology (unlike graph convolution networks, which require a fixed spatial topology to create an adjacency matrix), so I think this method could generalize well to body skeletons.

3. Can the visual representation handle missing data or occlusions robustly?

**Ethical Concerns:**

["NO or VERY MINOR ethics concerns only"]

**Final Justification:**

Thanks to the authors for addressing my concerns beyond my anticipations. I believe all my concerns have been addressed. Therefore, I've changed my score to accept.

**Limitations:**

yes

**Paper Formatting Concerns:**

There are no major formatting issues.

**Quality:**

3

**Strengths And Weaknesses:**

**Strengths**:

1. Original and Clever Formulation. The conversion of skeleton sequences into visual plots is both novel and intuitive. By leveraging pretrained vision backbones, the method sidesteps the need for complex temporal-spatial modeling architectures.

2. High Practical Impact. The method is highly relevant for real-time systems such as VR/AR, robotics, and HCI. Its streaming-friendly window-based architecture is well-suited for deployment in online gesture detection settings.

3. Strong Experiment Results. The method consistently outperforms prior art on two major gesture benchmarks across multiple metrics (DR, FP, JI, FPS). Ablations are provided for normalization, DRE, loss components, etc.

**Weaknesses**:

1. Potential Performance Bottlenecks in Latency-Sensitive Scenarios. Despite real-time FPS claims, the use of vision transformers on graph images may still be computationally heavier than lightweight GCNs or 1D CNNs, particularly for mobile applications.

2. No Formal Analysis of Robustness. Although SKETCH handles viewpoint and subject variation empirically, there is no robustness analysis under noise, occlusion, or missing joints—common in real-world sensors.

---

> ### Author Rebuttal · Authors · 2025-07-31
>
> Due to space constraints, the references use the same alphabetical identifiers as those listed at the bottom of Reviewer bPvY’s rebuttal, with the venue details also provided extensively by bPvY.
>
> **Weakness 1**
> - We appreciate the reviewer’s comment on the **computational cost** of using vision transformers on graph images. We agree that such models may be less optimal for mobile application scenarios due to their higher overhead compared to lightweight baselines like GCNs or 1D CNNs. Nonetheless, as shown in **Tabs. A and B**, we believe the accuracy improvements of hand gesture recognition our method offers—particularly in interpretability and recognition performance—provide meaningful advantages that justify its use within our framework.
> - Furthermore, although transformer-based models are more demanding, our framework **still achieves real-time FPS (≥ 30)** performance in practice. This suggests that, despite the increased cost, the method remains applicable in latency-sensitive settings while maintaining a favorable balance between accuracy and efficiency. We also recognize the need for further improving computational efficiency. As such, we plan to explore future directions such as architectural simplification and model compression techniques to better accommodate resource-constrained environments of a model.
>
> **Question 1**
> - We agree that efficiency is crucial for online recognition. To address this, we explored **lightweight alternatives** to ViT and Swin (Tabs. A and B). While convolution-based models are computationally efficient, they underperform compared to transformer-based models.
> - The reason we primarily utilized transformer-based models is their ability to capture long-range dependencies across the entire temporal sequence of the input images. In contrast, convolution-based models, which rely on local receptive fields and spatial inductive biases [a], are less effective at globally understanding the overall flow of joint kinematics in gesture sequences. As shown in our comparison **Tabs. A and B**, while lightweight CNN models are faster and more computationally efficient, they do not perform as well as ViT-based models in capturing the global motion dynamics essential for accurate gesture recognition.
>
>  - **Training and scalability analysis**: Many prior works require significantly more training epochs due to training from scratch and rely on multi-stream fusion frameworks, which independently train and infer multiple models. For example, BlockGCN [58] uses four streams and ProtoGCN [b] uses six. As shown in **Tabs. A and B**, traditional methods typically train for more than three times as many epochs compared to ours. "In contrast, our method uses pretrained backbones, converging within 20 epochs without multi-stream fusion, thus avoiding the added overhead.
> Furthermore, since our input image resolution remains fixed regardless of the number of input frames or joints, our model maintains constant computational load and throughput. In contrast, many prior approaches incur increasing complexity as frame count or joint count grows.
>
> **Table A: Results on SHREC’19: benchmarks vs. our ResNet-18/34 models (224 and 384 denotes input resolution $H=W$).**
> |Methods|DR↑|FP↓|Parameters (M)|FLOPs (G)|FPS|Epochs ($\times 10$)|
> |-|-|-|-:|-:|-:|-:|
> |MS-G3D [27]|0.69|0.25|6.34|3.26|33|16|
> |DSTA [40]|0.81|0.08|13.72|3.24|114|48|
> |DDNet [52]|0.82|0.10|1.80|0.02|455|60|
> |OO-dMVMT [8]|0.88|0.05|0.20|0.001|172|10|
> |ProtoGCN (joint) [b]|0.86|0.05|4.12|15.38|30|15|
> |SKETCH (ResNet-18-224)|0.84|0.20|12.00|5.36|345|2|
> |SKETCH (ResNet-34-224)|0.89|0.02|22.11|10.81|196|2|
> |SKETCH (ViT-Large-384)|0.90|0.03|303.94|174.82|57|2|
> |SKETCH (Swin-Small-224)|0.88|0.04|49.63|8.55|256|2|
> |SKETCH(Swin-Base-384)|0.91|0.03|87.52|44.66|110|2|
> |SKETCH(Swin-Large-384)|0.92|0.02|195.76|100.29|70|2|
>
> **Table B: Results on SHREC’22: benchmarks vs. our ResNet-18/34 models.**
> |Backbone|DR↑|FP↓|JI↑|Parameters (M)|FLOPs (G)|FPS|Epochs ($\times 10$)|
> |-|-:|-:|-:|-:|-:|-:|-:|
> |DG-STA [5]|0.51|0.32|0.40|0.27|0.11|238|30|
> |PSUMNet [43]|0.62|0.24|0.52|-|-|41|20|
> |MS-G3D [27]|0.68|0.21|0.57|6.36|2.62|34|16|
> |DSTA [40]|0.73|0.24|0.61|13.72|2.12|109|48|
> |DDNet [52]|0.88|0.16|0.78|1.80|0.01|455|60|
> |OO-dMVMT [8]|0.92|0.09|0.85|0.21|0.001|224|10|
> |SKETCH(ResNet-18-224)|0.87|0.04|0.84|11.53|5.36|345|2|
> |SKETCH(ResNet-34-224)|0.86|0.07|0.81|21.64|10.81|196|2|
> |SKETCH(Swin-Base-384)|0.91|0.06|0.86|87.05|44.66|110|2|
> |SKETCH(Swin-Large-384)|0.92|0.07|0.06|195.28|100.28|70|2|
>
>  - **Efficiency and throughput analysis**: We carefully profiled our method’s computational efficiency by measuring throughput, normalization time (preprocessing), and the time required for both sequential and parallel image transformation (i.e., rendering; **Tab. C**). All measurements were conducted under the same computational environment. As reported, the combined preprocessing and model inference time ensures real-time FPS performance in practice.
>
> **Table C: Inference time (ms) breakdown of our method for different backbones**
> |Backbone|Preprocessing|Sequential Rendering|Parallel Rendering|Model Inference|
> |-:|:-:|:-:|:-:|:-:|
> |ResNet-18-224|0.1|4.3|0.1|2.9|
> |ResNet-34-224|0.1|4.3|0.1|5.1|
> |ViT-Large-384|0.1|5.0|0.1|17.6|
> |Swin-Small-224|0.1|4.3|0.1|3.9|
> |Swin-Base-384|0.1|5.0|0.1|9.1|
> |Swin-Large-384|0.1|5.0|0.1|14.1|
>
> **Question 2**
> - **Evaluation on human body skeletons** and challenging offline benchmarks
> Although SKETCH is primarily intended for online hand gesture recognition, we deliberately evaluated it far beyond that setting—across on/offline scenarios, human‑action recognition tasks, and both 2D and 3D skeleton modalities. While our scores are only marginally below those of models specially tailored to each dataset, they nevertheless remain highly competitive. Please also bear in mind that, owing to the **short rebuttal window, we could not fully optimize hyper‑parameters for all three additional datasets**; we anticipate even stronger results after proper tuning. We compared SKETCH against benchmark methods specialized for each dataset.
>    - **NW‑UCLA** [c] – an offline, multi‑view 3D human‑action benchmark in which training and test sequences are captured from different camera viewpoints.
>    - **JHMDB** [e] – an offline 2D human‑action dataset composed of movie and web clips that feature camera motion, occlusion, and viewpoint shifts.
>    - **SHREC’17** [d] – an offline 3D hand‑gesture benchmark that requires distinguishing single‑finger configurations from whole‑hand shapes.
> - Although SKETCH is primarily designed for online hand gesture recognition, we deliberately evaluated it beyond that scope—across both online and offline settings, human action recognition tasks, and 2D/3D skeleton modalities. Despite not being tailored to each dataset, our method achieved competitive performance, only marginally below heavily optimized baselines (**Tabs. D, E, and F**). We fully agree with the reviewer’s insight that, unlike graph convolutional networks which require a fixed spatial topology, our topology-agnostic design enables SKETCH to generalize well to diverse skeleton structures, including full-body representations. This flexibility allows SKETCH to perform robustly across various skeleton modalities and camera viewpoints. We also kindly note that further hyperparameter tuning, which was constrained by the limited rebuttal period, is likely to yield even stronger results.
>
> **Table D: Results on the NW-UCLA dataset for offline human action recognition (refer to [58] for modality abbreviations).**
> |Methods|Publication|Modalities|Acc.|
> |-|:-|:-:|-:|
> |DC-GCN+ADG|ECCV'20|J+B+JM+BM|95.3|
> |CTR-GCN|ICCV'23|J+B+JM+BM|96.5|
> |BlockGCN|CVPR'24|J+B+JM+BM|96.9|
> |GAP|ICCV'23|J|94.0|
> |TD-GCN|IEEE TMM'23|J|94.8|
> |InfoGCN|CVPR'22|J|96.5|
> |BlockGCN|CVPR'24|J|95.5|
> |SKETCH (Ours)||J|95.5|
>
> **Table E: Results on the JHMDB dataset for offline 2D human action recognition.**
> |Methods|Acc.|
> |-|-|
> |DDNet|77.20|
> |TDNet|79.30|
> |MSG3D|77.52|
> |CTR-GCN|73.02|
> |X3D|79.14|
> |C3D|80.20|
> |4s-AGCN|77.82|
> |SKETCH (Ours)|78.40|
>
> **Table F: Results on the SHREC'17 dataset (14 gestures)**
> |Methods|Acc.|
> |:-|:-:|
> |Key-frameCNN|92.9|
> |ParallelCNN|93.3|
> |STA-Res-TCN|94.5|
> |MFANet|94.6|
> |DDNet|94.6|
> |SKETCH (Ours)|94.6|
>
> **Question 3 & Weakness 2**
> - We conducted robustness experiments on the SHREC’19 dataset by applying missing-data perturbations—augmentations not used during training—to evaluate model generalization.
>    - **Missing-joint robustness**: Simulated joint-sensor failure by randomly deleting joints and measuring performance.
>    - **Occlusion robustness**: Simulated partial marker occlusions by masking joint coordinates over time and joints, then measuring performance.
>     - **Test noise robustness**: We injected white Gaussian noise directly into the joint coordinates of the training data and evaluated the model on test sequences corrupted with the same noise.
> - As shown in **Tabs. G and H**, we evaluated DR and FP under these three perturbations. Despite the synthetic perturbations, SKETCH remained robust by capturing the spatio-temporal flow of joint kinematics, rather than relying on individual joint trajectories. This ability to model long-range dependencies enabled SKETCH to handle missing joints effectively, without the need for additional augmentations of training, demonstrating its robustness in challenging conditions.
>
> **Table G: Robustness of our method under various perturbations of missing joints**
> |Perturbations|Missing Rate|DR↑|FP↓|
> |-|-|-|-|
> |-|-|0.92|0.02|
> |Sequence|30%|0.91|0.02|
> |Sequence|50%|0.89|0.02|
> |Sequence|70%|0.85|0.02|
> |Sequence|90%|0.73|0.34|
> |Frame|30%|0.90|0.02|
> |Frame|50%|0.78|0.20|
> |Frame|70%|0.06|0.00|
>
> **Table H:  Robustness of our method under noise perturbations with varying noise rates (std.)**
> |Perturbations|Std.|DR↑|FP↓|
> |-|-|-|-|
> |Noise|0.02|0.89|0.04|
> |Noise|0.03|0.89|0.04|
> |Noise|0.05|0.76|0.07|

---

> > ### Comment · Reviewer_D9gv · 2025-08-08
> >
> > Thanks for your answers. All my concerns have been addressed. I've changed my score to accept

---

> > > ### Author Response · Authors · 2025-08-08
> > >
> > > Dear Reviewer,
> > >
> > > We sincerely thank the reviewer for the time, effort, and thoughtful feedback provided throughout the review process. We greatly appreciate the reviewer’s careful consideration of our rebuttal and follow-up clarifications, and we deeply appreciate the opportunity to address the concerns raised through additional analyses and experiments. Your constructive comments have been invaluable in guiding us to refine and strengthen the clarity, completeness, and overall quality of our work.
> > >
> > > Best regards,

---

> ### Author Response · Authors · 2025-08-07
>
> Dear Reviewer,
>
> Thank you very much for your thoughtful and constructive review, and for the interest you have shown in our work.
>
> We have carefully addressed all the concerns, limitations, and questions you kindly raised in your initial review. If there are any points you feel were not fully resolved, or if you have additional feedback, we would greatly appreciate it—as it would help us further improve the quality and clarity of our manuscript.
>
> Thank you once again for your valuable time and support.
>
> Best regards,

---

### Official Review · Reviewer_bPvY · 2025-07-03

**Clarity:** 4
**Significance:** 3
**Originality:** 3
**Rating:** 5
**Confidence:** 4

**Summary:**

This paper presents a pipeline for online skeleton-based hand gesture recognition. The authors observe that existing approaches often suffer from a “geometric shortcut,” stemming from Euclidean-space-based representations or from using hand-crafted features that are difficult to interpret and generalize. Inspired by ViTST, the paper introduces a graph plot-based representation, which effectively converts windowed skeleton coordinates into line charts that plot normalized local joint movement per axis. This representation is more interpretable and can be processed by any vision-based model. To address the information loss regarding global motion, the authors propose Dynamic Range Embedding, which embeds z-normalized axis movement across a window. Experiments on SHREC’19 and SHREC’22 demonstrate that the proposed method operates in real time and outperforms previous approaches in both detection rate and false positive rate.

**Questions:**

* Does the method use sliding windows or non-overlapping windows for segmenting the data?
* How does the method perform if vectorized representations are used directly instead of converting features into pixel-based plots? For example, what if the normalized $N_f \times J$ feature is fed directly, rather than through image plotting?
* What is the final resolution of the generated graph images?

**Ethical Concerns:**

["NO or VERY MINOR ethics concerns only"]

**Limitations:**

* As mentioned in the limitation section, information from preceding windows is not utilized. Future work could consider adopting video processing models to leverage longer temporal context.

**Paper Formatting Concerns:**

N.A.

**Quality:**

4

**Strengths And Weaknesses:**

Strengths

- The paper is exceptionally well-written and easy to follow—arguably the clearest paper I’ve read in this NeurIPS cycle.

- The idea of converting skeleton coordinates to graph charts is novel and interesting. The proposed method demonstrates strong effectiveness compared to previous hand-crafted features and other baselines.

- The authors provide thorough ablation studies and experiments on all components, demonstrating the effectiveness of the plotting strategy, the presented feature, and the normalization technique.

Weaknesses

- The paper does not systematically analyze how the proposed feature performs under global rotations or changes in camera viewpoint. It remains unclear how robust the method is in these scenarios.

- Both SHREC’19 and SHREC’22 are relatively simple datasets for gesture recognition by today’s standards. The paper could be further strengthened by evaluating on more challenging gesture-language datasets, such as BOT57M.

---

> ### Author Rebuttal · Authors · 2025-07-30
>
> **Weakness 1**
> - Thank you for this insightful comment. To address it, we have conducted additional experiments on two benchmarks involving **global rotations and challenging camera viewpoints**:
> 	- **Northwestern-UCLA (NW-UCLA) Multiview Action3D** [c], which provides offline multiview 3D skeletons for human action recognition under varied **global rotations**, where the testing dataset is captured from **camera viewpoints distinct** from those used in the training dataset.
> 	- **Joint annotated Human Motion DataBase (JHMDB)** [e], an offline human action dataset, which offers 2D skeletons extracted from movie and web clips and encompasses **challenges such as camera motion, occlusion, motion blur, and significant viewpoint variation**.
> - Although SKETCH is primarily designed for online hand gesture recognition, we deliberately evaluated it beyond that scope–across both online and offline settings, human action recognition tasks, and 2D/3D skeleton modalities. Despite not being tailored to each dataset, our method achieved competitive performance, only marginally below heavily optimized baselines (**Tabs. A and B**). We believe this is due to SKETCH’s topology-agnostic design, which, unlike GCN-based methods that rely on predefined adjacency structures, enables robust generalization across varied skeleton representations and camera viewpoints. We also kindly note that further hyperparameter tuning, which was constrained by the limited rebuttal period, is likely to yield even stronger results. We compared SKETCH against benchmark methods specialized for each dataset, as shown in **Tabs. A and B**.
>
> **Table A: Results on benchmark and proposed methods on the JHMDB dataset for offline 2D human action recognition.**
> |Methods|Venue|Acc.|
> |-|-|-:|
> |TwoBranchesLSTM|Sensors'22|61.24|
> |EHPI|ITSC'19|65.50|
> |JMRN|WACV'22|68.55|
> |DDNet|MMAsia'19|77.20|
> |TDNet|IET Comput. Vis.'22|79.30|
> |ST-GCN|AAAI'18|62.69|
> |AA-GCN|CVPR'18|67.46|
> |MSG3D|CVPR'20|77.52|
> |CTR-GCN|ICCV'21|73.02|
> |X3D|CVPR'22|79.14|
> |C3D|CVPR'22|80.20|
> |4s-AGCN|WACV'24|77.82|
> |SKETCH (Ours)||78.40|
>
> **Table B: Results on benchmark and proposed methods on the NW-UCLA dataset for offline multi-view 3D human action recognition (refer to [58] for modality abbreviations: J = Joint, B = Bone, JM = Joint Motion, and BM = Bone Motion).**
> |Methods|Venue|Modalities|Acc.|
> |-|-|:-:|-:|
> |EnsembleTS-LSTM|ICCV'17|J+B|89.2|
> |2S-AGC-LSTM|CVPR'19|J+B+JM+BM|93.3|
> |4S-Shift-GCN|CVPR'20|J+B+JM+BM|94.6|
> |DC-GCN+ADG|ECCV'20|J+B+JM+BM|95.3|
> |CTR-GCN|ICCV'23|J+B+JM+BM|96.5|
> |InfoGCN|CVPR'22|J+B+JM+BM|96.6|
> |BlockGCN|CVPR'24|J+B+JM+BM|96.9|
> |GAP|ICCV'23|J|94.0|
> |TD-GCN|IEEE Trans. Multimed'23|J|94.8|
> |InfoGCN|CVPR'22|J|96.5|
> |BlockGCN|CVPR'24|J|95.5|
> |SKETCH (Ours)||SKETCH|95.5|
>
> **Weakness 2**
> - We sincerely appreciate the reviewer’s valuable suggestion to evaluate our method on more challenging gesture-language datasets, such as BOT57M. As suggested, we have initiated the request for access to the BOT57M dataset through its official site; however, we have not yet received the data. Once access is granted, we plan to incorporate experiments on this dataset in our future revision to further validate the robustness and generalizability of SKETCH.
> In the meantime, to further demonstrate robustness beyond SHREC’19 and SHREC’22, we have also applied our approach to three additional benchmarks:
> In response to **Weakness 1** and alongside evaluations on the **NW-UCLA and JHMDB datasets**, as shown in **Tabs. C and D**, **SHREC'17** [d], an **offline hand‑gesture** benchmark requiring the distinction of single‑finger configurations from whole‑hand shapes.
>
> **Table C: Results on legacy benchmark methods–no recent baselines are available–on the SHREC’17 dataset (14 gestures) for offline hand gesture recognition.**
> |Methods|Venue|Acc.|
> |-|-|-:|
> |Dynamic Hand|CVPRW'16|88.2|
> |Key-frameCNN|3DOR'17|92.9|
> |3Cent|STAG'17|88.6|
> |CNN+LSTM|PR'18|91.8|
> |ParallelCNN|RFAI'18|93.3|
> |STA-Res-TCN|Gesture'18|94.5|
> |MFANet|Sensor'19|94.6|
> |DDNet|MMAsia'19|94.6|
> |SKETCH (Ours)||94.6|
>
>
> **Table D: Results on benchmark and proposed methods on SHREC'17 (28 gestures)**
> |Methods|Venue|Acc.|
> |-|:-:|-:|
> |PLSTM-base|CVPR'20|87.6|
> |PLSTM-early|CVPR'20|93.5|
> |PLSTM-PSS|CVPR'20|93.1|
> |PLSTM-middle|CVPR'20|94.7|
> |PLSTM-late|CVPR'20|93.5|
> |P4Transformer (30 Epochs)|CVPR'21|87.5|
> |P4Transformer (50 Epochs)|CVPR'21|91.2|
> |Kinet|CVPR'22|95.2|
> |PointCMP|CVPR'23|93.3|
> |P4Transformer+MaST-Pre (30 Epochs)|ICCV'23|90.2|
> |P4Transformer+MaST-Pre (50 Epochs)|ICCV'23|92.4|
> |SKETCH (Ours)||91.4|
>
> - Although **SKETCH was originally designed for online data and we were unable to perform full hyperparameter tuning due to the limited rebuttal period**, our method nonetheless delivers competitive performance on NW‑UCLA, JHMDB, and SHREC’17–falling only marginally below current state‑of‑the‑art results.
>
> **Question 1**
> - We apologize for any confusion caused. We acknowledge that the manuscript lacks a detailed description of this aspect. To clarify, the sliding window technique used in our experiments assumes a fixed window size of $W$, with a stride of 1, which effectively results in an overlap of $W-1$ frames. For better understanding, we kindly refer the reviewer to **Figs. 2 and 10 (Appendix F)**, which illustrates this window-based formulation for online gesture recognition. We will revise the manuscript to describe this setting more clearly and explicitly.
>
> **Question 2**
> - Thank you for the valuable question. We agree that directly using normalized $N_f \times J$ feature matrices–rather than converting them into pixel-based image plots–could serve as a useful comparison to isolate the impact of our SKETCH transformation.
> However, our backbone architectures are pretrained on image data and thus are not directly compatible with vectorized representations. To approximate the reviewer’s intent, we conducted two proxy experiments on SHREC’19 and SHREC’22 (**Tabs. E and F**):
> 	- (i) vs. (ii): We modified the input layer of OO-dMVMT [8] to accept raw joint sequences and compared its performance to the original version using hand-crafted features (**Supp. B**).
> 	- (iii) vs. (iv): We compared a variant using raw joint data with a vanilla transformer (architecturally similar to ViT) against our proposed method using SKETCH-transformed images with a pretrained Swin-T backbone.
>
> - Across both comparisons, models using raw or vectorized skeleton input ((i) and (iii)) struggled to learn high-level representations–often overfitting to geometric shortcuts or coordinate bias (**Appendix A**). Furthermore, t-SNE visualizations in **Supp. B** showed that SKETCH embeddings formed semantically coherent clusters, while raw-input versions collapsed into low-level geometric groupings. Additional comparisons with DeepGRU [9] on raw input (**Tab. 2**) further support this observation.
> These results empirically validate the SKETCH transformation’s role in producing a more structured and discriminative input space, thereby enhancing model effectiveness and generalization.
>
> **Table E: Impact of SKETCH Transformation vs. Raw Input on SHREC’19 Performance**
> |Index|Methods|Backbone|Raw|Hand-crafted|SKETCH|DR↑|FP↓|
> |-|-|:-:|:-:|:-:|:-:|-:|-:|
> |(i)|OO‑dMVMT [8]|-|✓|||0.32|0.13|
> |(ii)|OO‑dMVMT [8]|-||✓||0.88|0.05|
> |(iii)|Ours|Vanilla-T|✓|||0.41|0.14|
> |(iv)|Ours|Swin-T|||✓|0.92|0.02|
>
> **Table F: Impact of SKETCH Transformation vs. Raw Input on SHREC’22 Performance**
> |Index|Methods|Backbone|Raw|Hand-crafted|SKETCH|DR↑|FP↓|JI↑|
> |-|-|:-:|:-:|:-:|:-:|-:|-:|-:|
> |-|DeepGRU|-|✓|||0.26|0.25|0.21|
> |(i)|OO‑dMVMT [8]|-|✓|||0.71|0.13|0.64|
> |(ii)|OO‑dMVMT [8]|-||✓||0.92|0.09|0.85|
> |(iii)|Ours|Vanilla-T|✓|||0.68|0.16|0.61|
> |(iv)|Ours|Swin-T|||✓|0.92|0.07|0.87|
>
> **Question 3**
> - We acknowledge that the resolution details in the caption of **Tab. 1** may not have been as clear as intended, which could have caused some confusion for readers. To enhance clarity, we will revise the main text to state explicitly that the input spatial resolution is $224 \times 224$ for Swin Small and $384 \times 384$ for ViT Large, Swin Base, and ViT Large. In the additional experiments conducted during the rebuttal phase, the input size for ResNet-18 and ResNet-34 was also set to $224 \times 224$.
>
> **Limitations**
> - As mentioned in **Sec. 5.1 (Limitations and Future Work)**, we acknowledge that, as the reviewer pointed out, our current approach does not incorporate information exchange between previous windows, making it difficult to treat the model as one that fully understands the entire sequence. In future work, we plan to explore video and time-series processing methods [f] to address this limitation and improve our model's ability to capture longer temporal contexts.
>
> [a] Lou, Meng, et al. "OverLoCK: An Overview-first-Look-Closely-next ConvNet with Context-Mixing Dynamic Kernels." Proceedings of the Computer Vision and Pattern Recognition Conference. 2025.
>
> [b] Liu, Hongda, et al. "Revealing key details to see differences: A novel prototypical perspective for skeleton-based action recognition." Proceedings of the Computer Vision and Pattern Recognition Conference. 2025.
>
> [c] Wang, et al. "Cross-view action modeling, learning and recognition." Proceedings of the IEEE conference on computer vision and pattern recognition. 2014.
>
> [d] De Smedt, et al. "Shrec'17 track: 3d hand gesture recognition using a depth and skeletal dataset." 3DOR-10th Eurographics workshop on 3D object retrieval. 2017.
>
> [e] Jhuang, et al. "Towards understanding action recognition." Proceedings of the IEEE international conference on computer vision. 2013.
>
> [f] Pang, et al. "Context-enhanced memory-refined transformer for online action detection." Proceedings of the Computer Vision and Pattern Recognition Conference. 2025.

---

> ### Author Response · Authors · 2025-08-07
>
> Dear Reviewer,
>
> Thank you very much for your thoughtful and constructive review, and for the interest you have shown in our work.
>
> We have carefully addressed all the concerns, limitations, and questions you kindly raised in your initial review. If there are any points you feel were not fully resolved, or if you have additional feedback, we would greatly appreciate it—as it would help us further improve the quality and clarity of our manuscript.
>
> Thank you once again for your valuable time and support.
>
> Best regards,

---

### Official Review · Reviewer_JPRk · 2025-07-04

**Clarity:** 3
**Significance:** 3
**Originality:** 3
**Rating:** 4
**Confidence:** 3

**Summary:**

The paper introduces SKETCH (Skeleton Kinematics Extraction Through Coordinated grapH), a novel method that transforms 4D skeleton data (time, x, y, z) into a visually interpretable representation. This transformation enables the application of advanced visual recognition architectures such as SWIN and ViT for skeleton-based hand gesture recognition. The proposed approach seeks to make gesture recognition more intuitive and effective by converting temporal motion patterns into a visual domain.

While the visual idea is interesting and the method shows promise, the motivation lacks depth in positioning the contribution relative to the rich body of existing gesture recognition work. Further, model complexity details are absent, and the evaluation is limited to only two datasets, which restricts insight into the generalizability of the approach.

**Questions:**

Provide a more detailed discussion of what limitations exist in current gesture recognition methods and how SKETCH directly addresses them.
Explain why transformer-based models (SWIN, ViT) are necessary for this problem. Could lighter models suffice, especially given the reduced input complexity (keypoints instead of raw video)?
Report parameter count, FLOPs, and runtime comparisons to provide readers with a sense of computational efficiency and feasibility.
Consider evaluating on additional gesture datasets, especially those with varying motion complexity and user diversity, to assess the robustness and generalizability of SKETCH.

An ablation showing the impact of the SKETCH transformation versus raw skeleton input directly into the models would help isolate the contribution of your method.

**Ethical Concerns:**

["NO or VERY MINOR ethics concerns only"]

**Limitations:**

As discussed above

**Quality:**

3

**Strengths And Weaknesses:**

Transforming 4D skeleton data into a visual format is creative and facilitates the use of powerful visual transformer architectures.
The integration of vision transformer-based models (e.g., SWIN, ViT) for gesture recognition through the visual encoding is novel and bridges spatial-temporal modeling with image-based classification.
The SKETCH framework is conceptually aligned with the nature of the task (hand gestures), which makes the overall presentation engaging and easier to follow.

The paper does not sufficiently highlight the limitations of existing gesture recognition approaches or clearly justify why a transformation to a visual representation is necessary or superior.


Details such as the number of parameters, computational cost (GFLOPs), or inference time are not provided. These are essential when using large models like SWIN or ViT on a task typically solved with lighter architectures.


Given the task focuses on keypoint dynamics, the use of heavy models like ViT and SWIN-T seems excessive unless justified through clear comparative analysis or accuracy-performance trade-offs.


The experiments are conducted on only two datasets. Evaluating on more diverse and dynamic datasets would strengthen the generalizability claims.

---

> ### Author Rebuttal · Authors · 2025-07-30
>
> We will incorporate the additional comments into the paper. Due to space constraints, the references use the same alphabetical identifiers as those listed at the bottom of Reviewer bPvY’s rebuttal.
>
> **Limitations of Prior Skeleton Methods and SKETCH’s Solutions**
> - Prior work and limitations
>    - Skeleton-based gesture recognition has traditionally started from approaches that directly utilize **① raw joint coordinates**. However, this direction presents several challenges:
>      1. **Geometric shortcuts** [49]: Overfitting to low-dimensional cues like absolute coordinates and distances.
>      2. **Coordinate bias** [54]: Sensitivity to changes in viewpoint and scale.
>      3. **Inter subject differences**: Difficulty in generalizing due to variations in hand size and range of motion.
>    - To address these challenges, **② hand-crafted features** have been used [8, 58, b], but they face limitations:
>      4. **Expert knowledge**: Dependence on specific skeleton formats or datasets.
>      5. **Interpretability and raw data richness**: Limited interpretability regarding which joints or temporal segments influence decisions, and inability to fully exploit the richness of raw data.
>      6. **Local inductive biases and long‑range dependencies** [a]: 3D/4D CNNs methods [45] constrained by local inductive biases, limiting their ability to model long‑range dependencies across space and time.
>  - Why visualization is needed and how SKETCH addresses existing limitations
>    -  Addresses **① raw data** challenges:
>       1. By transforming raw coordinates into images, SKETCH uses pretrained visual backbones to extract high-level features, preventing overfitting to **geometric cues and reducing reliance on shortcuts**.
>       2. **Coordinate bias** is mitigated through the pretrained backbone’s invariance to spatial transformations, enabling gesture recognition despite changes in viewpoint, scale, or location.
>       3. **Inter-subject variability** is addressed through per-window normalization, using relative joint positions in image space.
>    -  Overcomes Limitations of **② Hand-Crafted Features**:
>       4. Our representation eliminates the need for **expert design**, adopting a transformation similar to how humans interpret visual charts, making it generalizable and intuitive.
>       5. SKETCH enables **interpretability** by converting raw skeletons into visual images, allowing inspection of joint and temporal segment contributions to classification (**Figs. 4, 7 (Appendix D) , and 14 (Appendix F)**).
>       6. Once mapped to the image domain, our data can be readily processed by global self-attention backbones, enabling explicit modeling of **long-range dependencies** between joints.
>  - Contributions of the work
>    - The proposed SKETCH method leverages raw skeleton data to enhance interpretability and preserve rich spatial-temporal information, resulting in improved recognition performance compared to prior approaches (**Tabs. 1 and 2**). Under these conditions, SKETCH shows fast convergence in small number of epochs ($≤ 20$ epochs) (**Tab. B**) and high generalization (**Tabs. C and D**).
>
> **Model Complexity and Efficiency**
>  - We conducted additional experiments on model complexity (**Tab. A**), which will be incorporated into the main paper. As noted, SKETCH’s computational cost was high with Transformer architecture backbones. To address this, we evaluated the method using lighter ResNet-18 and 34 architectures and plan to explore reducing computational costs through architectural simplification or model compression.
> Additionally, since our framework transforms each window  into a graph image of fixed spatial resolution, throughput and inference speed remain constant, unlike competing methods, where computational load increases with sequence  length or joint count.
>
> **Table A: Results of computational costs with state-of-the-art methods for action recognition.**
> |Methods|Venue|Params. (M)|FLOPs (G)|FPS|
> |-|-:|-:|-:|-:|
> |DC-GCN+ADG|ECCV'20|4.90|1.83|-|
> |EfficientGCN-B4|TPAMI'22|2.00|15.20|-|
> |MS-G3D|CVPR'20|6.36|3.26|33|
> |DSTA|ACCV'20|13.72|3.24|114|
> |MST-GCN|AAAI'21|12.00|-|-|
> |OO-dMVMT|CVPRW'23|0.20|0.001|172|
> |BlockGCN|CVPR'24|5.20|6.52|-|
> |ProtoGCN (joint)|CVPR'25|4.12|15.38|30|
> |SKETCH (ResNet-18)||12.00|5.36|345|
> |SKETCH (ResNet-34)||22.11|10.81|196|
> |SKETCH (ViT-Large)||303.94|174.82|57|
> |SKETCH (Swin-Small)||49.63|8.55|256|
> |SKETCH (Swin-Base)||87.52|44.66|110|
> |SKETCH (Swin-Large)||195.76|100.29|70|
>
> **Necessity of Transformers Compared to Lightweight Backbones**
>  - We understand the reviewer’s concern that using heavy models like ViT or Swin may be excessive for graph image input. To address this, we include a comparison (**Tab. B**) evaluating SKETCH with both lightweight CNN backbones and transformer models. As shown, while CNN backbones have lower computational costs, their performance lags behind transformer models.
> This gap is mainly due to ViT and Swin’s ability to capture long-range dependencies across the entire temporal image window, which is essential for understanding joint kinematics in gesture sequences (**Appendix E.1**). In contrast, CNNs rely on local receptive fields and inductive biases that may not fully capture global temporal-spatial patterns [a].
>
> **Table B: Results on SHREC’19: benchmarks vs. our ResNet models (224 and 384 denotes input resolution $H=W$).**
> |Methods|DR↑|FP↓|Params. (M)|FLOPs (G)|FPS|Epochs ($\times$10)|
> |-|-:|-:|-:|-:|-:|-:|
> |DSTA [40]|0.81|0.08|13.72|3.24|114|48|
> |DG-STA [5]|0.81|0.07|0.27|0.17|238|30|
> |OO-dMVMT [8]|0.88|0.05|0.20|0.001|172|10|
> |ProtoGCN (joint) [b]|0.86|0.05|4.12|15.38|30|15|
> |SKETCH (ResNet-18-224)|0.84|0.20|12.00|5.36|345|2|
> |SKETCH (ResNet-34-224)|0.89|0.02|22.11|10.81|196|2|
> |SKETCH (ViT-Large-384)|0.90|0.03|303.94|174.82|57|2|
> |SKETCH (Swin-Small-224)|0.88|0.04|49.63|8.55|256|2|
> |SKETCH (Swin-Base-384)|0.91|0.03|87.52|44.66|110|2|
> |SKETCH (Swin-Large-384)|0.92|0.02|195.76|100.29|70|2|
>
> **Why Raw Keypoints Should Not Be Used Directly**
>  - We appreciate the valuable comment. Directly using raw keypoint coordinates introduces challenges such as geometric shortcut and coordinate bias, which hinder generalization and semantic learning. The analysis is described in **Appendix A**, **Supp. B**, and the additional experiments in **Tab. E**. To address this, we proposed SKETCH as an intermediate medium that transforms raw keypoints into structured images—making it feasible to leverage pretrained vision backbones.
>
> **Evaluation on the Generalization and Robustness**
>  - Cross-dataset evaluation on challenging offline benchmarks:
>    - Although SKETCH is designed for **online hand-gesture recognition**, we evaluated it across on/offline scenarios and human-action recognition tasks. While our results are slightly below those of models specifically tailored to each dataset, they remain competitive. Note that due to the short rebuttal period, we couldn't fully optimize hyperparameters for all three datasets, but we expect stronger results after further tuning. We compared SKETCH against benchmark methods specialized for each dataset, as shown in **Tabs. C and D**.
>       - **NW-UCLA** [c]: An **offline multi-view 3D human-action** benchmark with sequences captured from **different camera viewpoints**.
>       - **SHREC’17** [d]: An **offline 3D hand-gesture** benchmark requiring distinction between single-finger configurations and whole-hand shapes.
>
> **Table C: Results on the NW-UCLA dataset for offline human action recognition (refer to [58] for modality abbreviations, J+B+JM+BM=M).**
> |Methods|Venue|Modalities|Acc.|
> |-|-|:-:|-:|
> |4S-Shift-GCN|CVPR'20|M|94.6|
> |DC-GCN+ADG|ECCV'20|M|95.3|
> |CTR-GCN|ICCV'23|M|96.5|
> |InfoGCN|CVPR'22|M|96.6|
> |BlockGCN|CVPR'24|M|96.9|
> |GAP|ICCV'23|J|94.0|
> |TD-GCN|IEEE Trans. Multimed.'23|J|94.8|
> |BlockGCN|CVPR'24|J|95.5|
> |SKETCH (Ours)||SKETCH|95.5|
>
> **Table D: Results on legacy benchmark methods—no recent baselines are available—on the SHREC’17 dataset (14 gestures) for offline hand gesture recognition.**
> |Methods|Venue|Acc.|
> |-|-|:-:|
> |CNN+LSTM|PR'18|91.8|
> |ParallelCNN|RFAI'18|93.3|
> |STA-Res-TCN|Gesture'18|94.5|
> |MFANet|Sensor'19|94.6|
> |DDNet|MMAsia'19|94.6|
> |SKETCH (Ours)||94.6|
>
> **Ablation Study of Sketch versus Raw Skeleton Inputs to the Model**
> - As the reviewer noted, isolating the contribution of the SKETCH transformation ideally requires comparing the same backbone architecture with and without SKETCH. However, our vision transformer backbones are pretrained on image inputs and cannot directly accept raw skeleton coordinates, and methods like OO-dMVMT [8] are designed for hand-crafted features and are incompatible with raw joint inputs due to dimensional mismatches.
> - To address this, we conducted a **proxy comparison** using SHREC’19 as benchmarks (**Tab. E**):
>    - (i) vs. (ii): We modified OO-dMVMT to accept raw skeleton data and compared it to the original version with hand-crafted features (**Supp. B**).
>    - (iii) vs. (iv): We compared a vanilla transformer (similar to ViT) with raw inputs against our SKETCH-transformed input fed into a pretrained Swin-T backbone.
>
> - In these comparisons, raw-input variants ((i) and (iii)) failed to learn meaningful representations due to coordinate bias and geometric shortcuts (**Appendix A**). t-SNE visualizations in **Supp. B** show that SKETCH embeddings form semantically coherent clusters, while raw inputs collapsed into low-level groupings. Further comparisons with DeepGRU [9] on raw input (**Tab. 2**) support this observation.
>
> - These results empirically validate SKETCH’s role in generating a more structured and discriminative input space, improving model effectiveness and generalization.
>
> **Table E: Impact of SKETCH Transformation vs. Raw Input on SHREC’19 Performance**
> |Index|Methods|Backbone|Raw|Hand-crafted|SKETCH|DR↑|FP↓|
> |-|-|:-:|:-:|:-:|:-:|-:|-:|
> |(i)|OO‑dMVMT|-|✓|||0.32|0.13|
> |(ii)|OO‑dMVMT|-||✓||0.88|0.05|
> |(iii)|Ours|Vanilla-T|✓|||0.41|0.14|
> |(iv)|Ours|Swin-T|||✓|0.92|0.02|

---

> ### Author Response · Authors · 2025-08-05
>
> Dear Reviewer,
>
> We sincerely apologize for not providing the answer you were looking for. During the NeurIPS rebuttal process, we conducted additional experiments using the latest comparative method, CVPR 2025 ProtoGCN [b]. We will now add some more recent techniques and compare them with our proposed technique.
> For essential experiments, if you could suggest any essential experiments, we will incorporate them immediately. Moreover, we will reflect all work carried out during the rebuttal in the paper.
>
> Best regards,
>
> [b] Liu, Hongda, et al. "Revealing key details to see differences: A novel prototypical perspective for skeleton-based action recognition." Proceedings of the Computer Vision and Pattern Recognition Conference. 2025.

---

> > ### Author Response · Authors · 2025-08-06
> >
> > Dear Reviewer,
> >
> > Thank you for your valuable feedback. We sincerely appreciate the time and effort you have devoted to reviewing our work.
> >
> > We have endeavored to include a comprehensive set of experiments, as detailed in the Appendix and Supplementary Materials. However, we fully acknowledge that some key evaluations may not have fully met expectations. If you would be willing to clarify which specific “essential experiments” or recent comparative studies you feel are missing, we would greatly appreciate it and will prioritize addressing them thoroughly in the revised manuscript.
> >
> > Thank you once again for your thoughtful and constructive comments.

---

> ### Author Response · Authors · 2025-08-07
>
> **Response to the Omission of Recent Comparative Studies**
>
> Thank you for your careful review and honest feedback. We apologize for the omissions in our original submission. In response, we have optimized and integrated the latest methods [l, b, 58]—including those presented at AAAI 2024, CVPR 2024, and CVPR 2025—and added their results to **Tab. F**. We hope these substantial revisions address your concerns and strengthen our manuscript.
>
> **Table F: Comparison of SKETCH with recent state-of-the-art methods on SHREC’19. Values in parentheses indicate
> backbone configurations, where backbone type: V = ViT, S = Swin; model size: L = Large, B = Base, S = Small; patch size; Swin window size (for Swin only); and image resolution.**
> | Method| DR↑| FP↓|
> |-|-|-|
> | PSUMNET [43] | 0.64 | 0.22 |
> | MS-G3D [27]| 0.69 | 0.25 |
> | SeS-GCN [37]| 0.75 | 0.12 |
> | SW 3-cent [3]| 0.76 | 0.19 |
> | DSTA [40]| 0.81 | 0.08 |
> | DG-STA [5]| 0.81 | 0.07 |
> | DDNet [52]| 0.82 | 0.10 |
> | uDeepGRU [4] | 0.85 | 0.10 |
> | OO-dMVMT [8]  | 0.88 | 0.05 |
> | **DS-GCN (joint)** [l] | 0.80|0.05|
> |**BlockGCN (joint)** [58]|0.83|0.04|
> | **ProtoGCN (joint)** [b]|0.86|0.05|
> | SKETCH (V-L-16-384) | 0.90 | 0.03 |
> | SKETCH (S-S-4-7-224) | 0.88 | 0.04 |
> | SKETCH (S-B-4-12-384)| 0.91 | 0.03 |
> | **SKETCH (S-L-16-384)** | **0.92** | **0.02** |
>
> ---
>
> [b] Liu, Hongda, et al. "Revealing key details to see differences: A novel prototypical perspective for skeleton-based action recognition." Proceedings of the Computer Vision and Pattern Recognition Conference. **2025**.
>
> [I] Xie, Jianyang, et al. "Dynamic semantic-based spatial graph convolution network for skeleton-based human action recognition." Proceedings of the AAAI conference on artificial intelligence. Vol. 38. No. 6. **2024**.

---

> ### Author Response · Authors · 2025-08-07
>
> **Response to Unaddressed Essential Experiments**
>
> In light of your helpful comments, we revisited the experimental design and carefully considered possible areas where further validation might strengthen the manuscript. Motivated by your emphasis on essential experiments, we conducted a new set of evaluations that explore the effects and limitations of different normalization strategies, their applicability across diverse architectures, and the benefits of our proposed dynamic range embedding. We hope these studies clarify concerns and show our method’s strengths.
>
> **1. Limitations of Naïve Normalization**
>
> As shown in **Tab. G**, training a vanilla Transformer model on raw skeleton sequences using **naïve normalization (NN)** results in notably limited performance. This setting suffers from **geometric shortcuts and coordinate bias**, leading the model to overfit to low-level geometric patterns that do not generalize well across subjects or viewpoints.
>
> **Table G: Performance of vanilla Transformer with NN on SHREC’19 and SHREC’22.**
> |Dataset|DR↑|FP↓|JI↑|
> |:-:|:-:|:-:|:-:|
> |SHREC'19|0.41|0.14|-|
> |SHREC'22|0.68|0.16|0.61|
>
> ---
> **2. Effectiveness and Limitations of Analytical Normalisation**
>
> To empirically validate the benefits of view- and subject-invariant **AN**, we compared our method with existing approaches that explicitly adopt such strategies, including **BlockGCN** [58], **DS-GCN** [l], and **ProtoGCN** [b], on the SHREC’19 dataset (**Tab. H (i-iii)**).
>
>   - **BlockGCN**, **DS-GCN**, and **ProtoGCN**: Implements viewpoint invariance by converting all joints to relative coordinates based on a reference joint.
>
> These methods demonstrate the **effectiveness of AN**, and our experiments confirm that such strategies alleviate view- and subject-dependency as the Reviewer noted.
>
> However, we also emphasise that **AN may introduce its own limitations**:
>
>   - It often **requires domain expert knowledge**, such as selecting an appropriate reference joint based on prior topology information for a given dataset or sensor configuration.
>   - The definition and index of the **reference joint may differ across datasets** (e.g., SpineBase at index 0 in Kinect V2, MidHip at index 8 in OpenPose BODY-25) or **even be absent** (e.g., COCO 2D lacks a central reference joint).
>   - As the Reviewer highlighted, **real-world scenarios are prone to pose estimation errors**; if the reference joint is noisy or missing, **AN** can severely degrade performance due to its reliance on that joint.
>
> In contrast, our proposed SKETCH method is **topology-agnostic**—it does not rely on any specific reference joint or domain-specific preconditions —which enables **generalisation across diverse skeleton structures and provides resilience to pose estimation errors**, as it avoids dependencies that may be fragile in real-world scenarios with missing or noisy joints.
>
> **Table H: Performance comparison on SHREC’19 between SKETCH and methods employing AN.**
> |Index|Dataset|Method|Venue|AN|DR↑|FP↓|
> |-|:-:|:-:|:-:|:-:|:-:|:-:|
> |(i)|SHREC'19|BlockGCN|CVPR'24|✓|0.83|0.04|
> |(ii)|SHREC'19|DS-GCN|AAAI'24|✓|0.80|0.05|
> |(iii)|SHREC'19|ProtoGCN|CVPR'25|✓|0.86|0.05|
> |(iv)|SHREC'19|SKETCH|||0.92|0.02|
>
> ---
> **3. Analytical Normalization with Vanilla Transformer Backbone**
>
> To further isolate the effects of normalization, we applied **AN** schemes to a **vanilla Transformer backbone**, which is architecturally similar to SKETCH (without the image-based transformation). Results on SHREC’19 and SHREC’22 (**Tab. I (i) and (iii)**) confirm that:
>   - **AN** aimed at addressing view-dependence shows clear performance improvements even under this experimental setting.
>
> **Table I: Ablation study on SHREC’19 and SHREC’22 under AN to evaluate the performance impact of DRE.**
> |Index|Dataset|AN|DRE|DR↑|FP↓|JI↑|
> |-|:-:|:-:|:-:|:-:|:-:|:-:|
> |(i)|SHREC'19|✓||0.62|0.09|-|
> |(ii)|SHREC'19|✓|✓|0.64|0.09|-|
> |(iii)|SHREC'22|✓||0.81|0.15|0.72|
> |(iv)|SHREC'22|✓|✓|0.82|0.13|0.75|
>
> ---
> **4. Compatibility of DRE with Analytical Normalization**
>
> We introduced **Dynamic Range Embedding (DRE)** to recover motion magnitude information lost during SKETCH’s axis-wise min–max normalization. By learning and injecting the dynamic range of each axis into the visual representation, DRE helps retain scale-related cues important for distinguishing gestures. While a direct comparison between **DRE** and **AN** is not straightforward due to their differing objectives, we conducted a **proxy experiment** by applying **DRE** on top of **AN**—following the protocol used in **BlockGCN** [58]—within a vanilla Transformer backbone, and evaluated its impact on gesture recognition.
>
>   - See **Tab. I (i–ii)** for SHREC’19 results without and with **DRE**, and **Tab. I (iii-iv)** for corresponding results on SHREC’22.
>
> These results indicate that **DRE** provides complementary benefits even when **AN** is applied, by restoring axis-specific magnitude cues lost in the normalization process.

---

> ### Author Response · Authors · 2025-08-07
>
> If there are any additional experiments or comparative evaluations you believe would strengthen the work, we would greatly appreciate your guidance. We are fully committed to incorporating further revisions to meet your expectations and improve the quality and completeness of our work.
>
> Best regards,

---

### Note · Authors · 2025-08-13

We sincerely thank all Reviewers and the Area Chair for their time and constructive feedback, which have significantly contributed to improving the quality and clarity of our work.

Through the rebuttal and discussion process, we have identified several aspects to strengthen in the revised manuscript:

---
**Topology-agnostic design**
- Further emphasized SKETCH’s topology-agnostic nature, showing consistent performance across skeleton types, datasets, and 2D/3D modalities, highlighting broad generalization.

**[Key findings]**
- SKETCH adapts seamlessly to varying skeleton configurations without structural priors.
- Maintains accuracy and robustness even with missing joints or occlusions, confirmed across multiple datasets and settings.

→ This flexibility strengthens the rationale and applicability of our approach across domains.

---
**Computational efficiency metrics**
- Added quantitative complexity indicators (parameter count, FLOPs, FPS, latency) to tables alongside accuracy metrics.

**[Key findings]**
- SKETCH maintains real-time performance (≥ 30 FPS) even with transformer-based backbones.
- Lightweight CNN backbones offer higher FPS but lower accuracy, justifying the selected trade-off.
---
**Interpretability statements**
- Refined statements to clearly distinguish between interpretability inherent to skeletal inputs and the transparency afforded by our visual representation.

**[Key findings]**
- SKETCH preserves joint-level semantics in visual form, enabling intuitive attention inspection while avoiding geometric shortcuts.
---
**Analytical Normalization (AN) and Dynamic Range Embedding (DRE)**
- Evaluated AN via comparisons with representative AN-based methods, and conducted proxy studies applying DRE on top of AN to examine complementary effects.

**[Key findings]**
- AN alleviates view- and subject-dependence but can be fragile when reference joints are noisy or missing.
- DRE complements AN by restoring axis-specific magnitude cues lost during normalization.

→ These results better contextualize SKETCH’s advantages over both AN-based and topology-dependent approaches.

---
**Methodological clarifications**
- Provided explicit details on sliding window configuration, input resolutions for different backbones.
---
We are committed to incorporating all these improvements in the revised version, and we deeply respect and acknowledge the dedication of the AC and Reviewers, whose efforts reflect a genuine commitment to advancing AI research.

---

### Decision · Program_Chairs · 2025-09-17

**Decision:**

Accept (poster)

**Comment:**

The paper proposes SKETCH – a lightweight pipeline that converts 4‑D skeleton streams into a compact, interpretable visual representation and feeds the resulting images to pre‑trained vision transformers (ViT/Swin). By embedding a Dynamic Range Embedding (DRE) to preserve relative motion scale, the method achieves real‑time, state‑of‑the‑art gesture detection on the SHREC’19/22 benchmarks while remaining highly interpretable.
In the rebuttal, the authors have successfully addressed most of the reviewers' concerns, resulting in two "accept" and two "borderline accept" ratings for the paper.
Given the strong novelty, practical relevance, and the authors’ thorough responses to reviewer concerns, we recommend acceptance of the paper. The contributions are clear, the experiments robust, and the method has immediate applicability to real‑world gesture‑recognition systems.